# Quantum-coherent mixtures of causal relations

Jean-Philippe W. MacLean[1,2,*], Katja Ried[1,2,3,*], Robert W. Spekkens[3] & Kevin J. Resch[1,2]

Understanding the causal influences that hold among parts of a system is critical both to explaining that system's natural behaviour and to controlling it through targeted interventions. In a quantum world, understanding causal relations is equally important, but the set of possibilities is far richer. The two basic ways in which a pair of time-ordered quantum systems may be causally related are by a cause-effect mechanism or by a common-cause acting on both. Here we show a coherent mixture of these two possibilities. We realize this nonclassical causal relation in a quantum optics experiment and derive a set of criteria for witnessing the coherence based on a quantum version of Berkson's effect, whereby two independent causes can become correlated on observation of their common effect. The interplay of causality and quantum theory lies at the heart of challenging foundational puzzles, including Bell's theorem and the search for quantum gravity.

[1] Institute for Quantum Computing, University of Waterloo, Waterloo, Ontario, Canada N2L 3G1. [2] Department of Physics & Astronomy, University of Waterloo, Waterloo, Ontario, Canada N2L 3G1. [3] Perimeter Institute for Theoretical Physics, Waterloo, Ontario, Canada N2L 2Y5. * These authors contributed equally to this work. Correspondence and requests for materials should be addressed to J.-P.W.M. (email: jpmaclean@uwaterloo.ca) or to K.R. (email: katja.ried@uibk.ac.at).

Unravelling the causal mechanisms that explain observed correlations is an important problem in any field that uses statistical data. For example, a positive correlation between the damage done by a fire and the number of firefighters on scene does not imply that the firefighters caused the damage. Discovering causal relations has applications ranging from epidemiology and genetics to economics and policy analysis[1,2]. Causal explanation is also playing an increasingly important role in quantum physics. It has recently gained prominence in the analysis of Bell's theorem and generalizations thereof[3–6]. Furthermore, causal structure is a close proxy for the structure of space-time in general relativity, and it has been suggested that we will have to abandon the notion of definite causal structure and instead allow superpositions thereof to develop a theory of quantum gravity[7,8]. Understanding causality in a quantum world may provide new resources for future quantum technologies as we gain control over increasingly complex quantum systems. For instance, entanglement has been shown to provide a quantum advantage for causal inference in certain causal scenarios[9].

Classically, when two time-ordered variables are found to be statistically correlated, there are different causal mechanisms that could explain this. It could be that the early variable causally influences the later one, or that both are effects of a common cause. Alternatively, the relation could be either cause-effect or common-cause with certain probabilities. Most generally, there may be cause-effect and common-cause mechanisms acting simultaneously. We refer to this as a physical (as opposed to probabilistic) mixture of the two mechanisms.

In a quantum world, there are additional possibilities. Purely common-cause mechanisms are intrinsically quantum if they correspond to entangled bipartite states. Purely cause-effect mechanisms are intrinsically quantum if they correspond to channels that are not entanglement breaking. But quantum effects are not restricted to only these: as we will show, when common-cause and cause-effect mechanisms act simultaneously, one can have quantum-coherent mixtures of causal relations. While conventional quantum mechanics can describe purely cause-effect and purely common-cause relations, quantum-coherent mixtures can only be represented using recent extensions of the formalism[10–16], in particular refs 9,17.

Chiribella[18] and Oreshkov et al.[14] have investigated coherent combinations of different causal orderings, specifically of $A$ causing $B$ and $B$ causing $A$. If realizable, such combinations would constitute a resource for computational tasks[19], with striking applications in gate discrimination[18,20,21]. However, this possibility requires that $A$ and $B$ are not embeddable into a global causal order, whereas current physical theories implicitly assume such an ordering. By contrast, we study a coherent combination of causal structures wherein $A$ is always temporally before $B$, a situation that is compatible with a global ordering and which therefore can be realized experimentally. If the demonstrated applications of superpositions of causal orders noted above can be attributed to the novel possibilities that are allowed by quantum theory for combining causal relations, then other quantum-coherent mixtures of causal relations, such as cause-effect and common-cause, may also constitute a resource.

The present work provides a framework for describing the different ways in which causal relations may be combined and experimental schemes for realizing and detecting them. We perform an experiment with photonic qubits that implements various such combinations and observes their operational signatures. Our main result is the experimental confirmation of the possibility of preparing a quantum-coherent mixture of common-cause and cause-effect relations.

## Results

**Signatures of different causal mixtures.** We seek to classify the causal relations that can hold between two quantum systems and, in particular, to derive and detect an experimental signature of a quantum-coherent mixture of cause-effect and common-cause relations. The tools for this can be illustrated with a simpler example: a mixture of two cause-effect mechanisms in a scenario wherein two distinct causes influence a common effect.

A signature for distinguishing physical from probabilistic mixtures can be derived from Berkson's effect, a phenomenon in classical statistics whereby conditioning on a variable induces statistical correlations between its causal parents when they are otherwise uncorrelated. Figure 1a,b provides an intuitive example. Note that the Berkson effect only arises if one has a combination of two causal mechanisms: in Fig. 1, both teaching and research ability influence the hiring decision. Crucially, the strength of the induced correlations can reveal how the two mechanisms are combined. In particular, probabilistic mixtures can only induce relatively weak correlations, as illustrated in Fig. 1c and proved rigorously in Supplementary Note 6. Correlations that are stronger than this bound bear witness to a physical mixture of causal mechanisms.

Quantum systems also exhibit the Berkson effect, with the strength of the induced correlations allowing one to distinguish physical from probabilistic mixtures. However, the generalization to quantum systems adds a third category to this classification: post selection may generate not just classical correlations, but quantum correlations (for example, entanglement) between the causal parents. We propose that this is a defining feature of a quantum-coherent mixture of causal mechanisms, a full definition of which will be developed below.

**Causal relations between two classical systems.** We now turn to the different causal relations between a pair of systems, labelled $A$ and $B$, with $A$ preceding $B$ in time. We begin by discussing the case of classical variables, which will motivate our definitions for the quantum case. Figure 2a depicts the paradigm example, a drug trial, and Fig. 2b–d introduces useful representations of the possible causal relations.

Both the example of a randomized drug trial (Fig. 2a, right) and the circuit representation (Fig. 2d, middle) show that a complete description of the causal relation between $A$ and $B$ involves two versions of the variable $A$: the version before the randomizing intervention, denoted $C$ (treatment preference), has a purely common-cause relation to $B$, whereas the post-intervention version, $D$ (assigned treatment), directly influences $B$. The causal relation between $A$ and $B$ is therefore completely specified by the stochastic map $P(CB|D)$.

This scenario supports a more general version of the Berkson effect: conditioning on recovery can induce correlations not only between the assigned treatment and the unobserved common cause, but also, by extension, between assigned treatment and treatment preference. These correlations bear witness to a combination of common-cause and cause-effect mechanisms, and their strength, as before, can distinguish different classes of combinations.

Before we develop a mathematical representation of these correlations in the quantum case, we first highlight a subtlety of the scenario by appealing to the classical case. In a randomized drug trial, the assigned treatment is controlled by the experimenter, hence there is no prior distribution over this variable. The object that encodes how assigned treatment correlates with treatment preference in the subpopulation that recovered is therefore not a joint distribution, but a map from assigned treatment to treatment preference. Given the overall stochastic map $P(CB|D)$, the subpopulation with $B = b$ is described by the

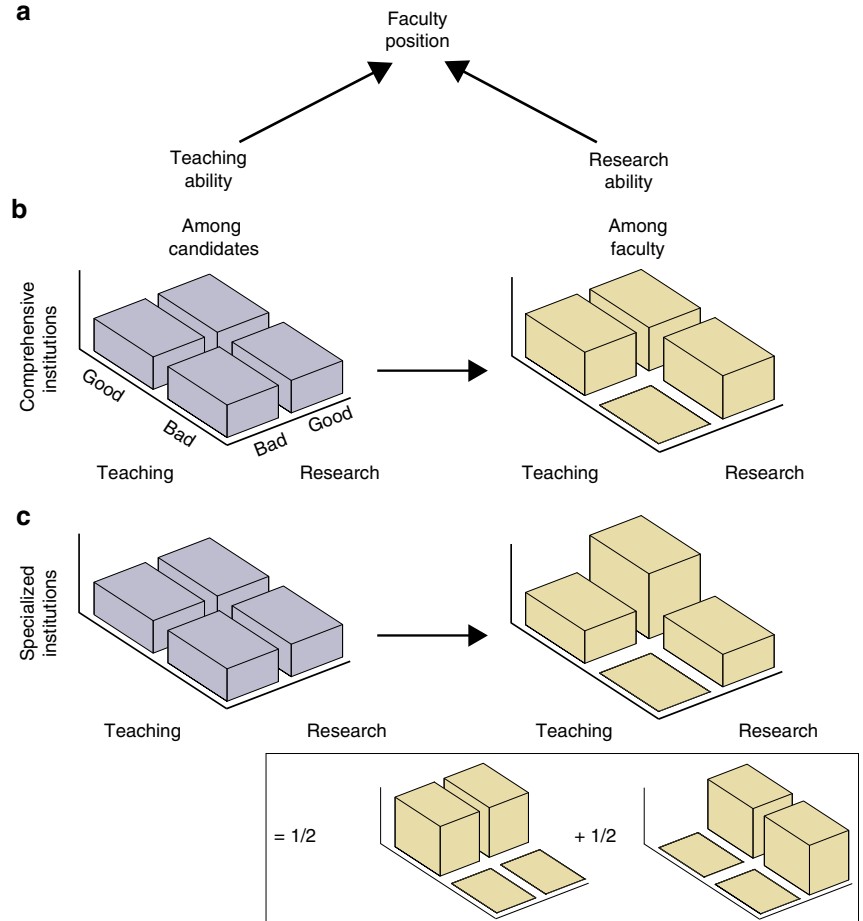

**Figure 1 | An illustration of Berkson's effect when hiring faculty in different institutions.** (**a**) When applying for faculty positions, a candidate's success generally depends on their skills at both teaching and research. We assume that these abilities are statistically independent in the overall field of applicants. (**b**) At comprehensive institutions, the hiring process considers both skills and eliminates candidates who are both bad teachers and bad researchers. Consequently, the two abilities become negatively correlated among successful candidates. (**c**) A set of specialized institutions, each one dedicated either purely to teaching or purely to research, select faculty based solely on the relevant ability in each case—a probabilistic mixture of both causal mechanisms as shown in the inset. Knowing that a candidate was successful in this scenario only reveals information about one of their abilities, and consequently induces weaker negative correlations than in **b**, due to the larger fraction of faculty members who are skilled at both.

element $P(C,B=b|D)$, which is a subnormalized stochastic map. If one wishes to quantify the correlations encoded in this map using standard measures, one can use the following prescription to construct a joint distribution that is isomorphic to $P(C,B=b|D)$: let $u(D)$ denote the uniform distribution over $D$ and take $P^b(CD) \equiv P(C, B=b|D)u(D)/P_b$, where $P_b \equiv \sum_{CD} P(C,B=b|D)u(D)$ is a normalization factor. This object encodes the correlations we wish to study in a convenient form and, moreover, admits a close quantum analogue, as we will show.

**Causal relations between two quantum systems.** If $A$ and $B$ are quantum systems, the input–output functionality of the circuits in Fig. 2 can be characterized using measurements on $B$ and an analogue of a randomized intervention on $A$, that is, a measurement followed by a random repreparation. As in the classical case, we split $A$ into $C$ and $D$. Mathematically, the circuit's functionality is represented by a trace-preserving, completely positive map from states on $D$ to states on the composite $CB$, $\mathcal{E}_{CB|D} : \mathcal{L}(\mathcal{H}_D) \to \mathcal{L}(\mathcal{H}_C \otimes \mathcal{H}_B)$, (where $\mathcal{L}(\mathcal{H}_X)$ denotes the linear operators over the Hilbert space of $X$), as can be inferred from refs 9,12–14,16, and which we term a causal map.

The Berkson effect on quantum systems is formalized as follows: consider a measurement on $B$, whose outcomes $b$ are

associated with positive operators $\{\Pi_B^b\}$. Finding an outcome $b$ implies correlations between $C$ and $D$, which are represented by a trace-non-increasing map from $D$ to $C$: $\mathcal{E}_{C|D}^b \equiv \mathrm{Tr}_B(\Pi_B^b \mathcal{E}_{CB|D})$ (analogous to the subnormalized stochastic map $P(C,B=b|D)$). Equivalently, we can represent this map using the quantum state $\tau_{CD}^b$ that one obtains by taking the operator that is Choi-isomorphic[22] to $\mathcal{E}_{C|D}^b$ and normalizing it to have unit trace (analogous to the normalized distribution $P^b(CD)$). The correlations between $C$ and $D$ embodied in the map $\mathcal{E}_{C|D}^b$ can then be assessed using standard measures of correlation on the state $\tau_{CD}^b$. We say that the causal map exhibits a quantum Berkson effect if there exists a measurement $\{\Pi_B^b\}$ such that for every outcome $b$, the induced correlations between $C$ and $D$, described by $\tau_{CD}^b$, are quantum. For the purposes of this article, we take the presence of entanglement as a sufficient condition for quantumness. Thus, our condition is that each $\tau_{CD}^b$ be entangled, or equivalently, that each $\mathcal{E}_{C|D}^b$ be non-entanglement breaking. Using these definitions, we will now propose a classification of the possible causal relations between two quantum systems, as well as ways of distinguishing the classes.

A causal map $\mathcal{E}_{CB|D}$ is purely cause-effect if it has the form $\mathcal{E}_{CB|D}(\cdot)=\mathcal{E}_{B|D}(\cdot) \otimes \rho_C$ (the analogue of $P(CB|D)=P(B|D)P(C)$), which makes it compatible with the causal structure in Fig. 2b; and purely common-cause if $\mathcal{E}_{CB|D}(\cdot)=\rho_{CB}\mathrm{Tr}_D(\cdot)$ (the

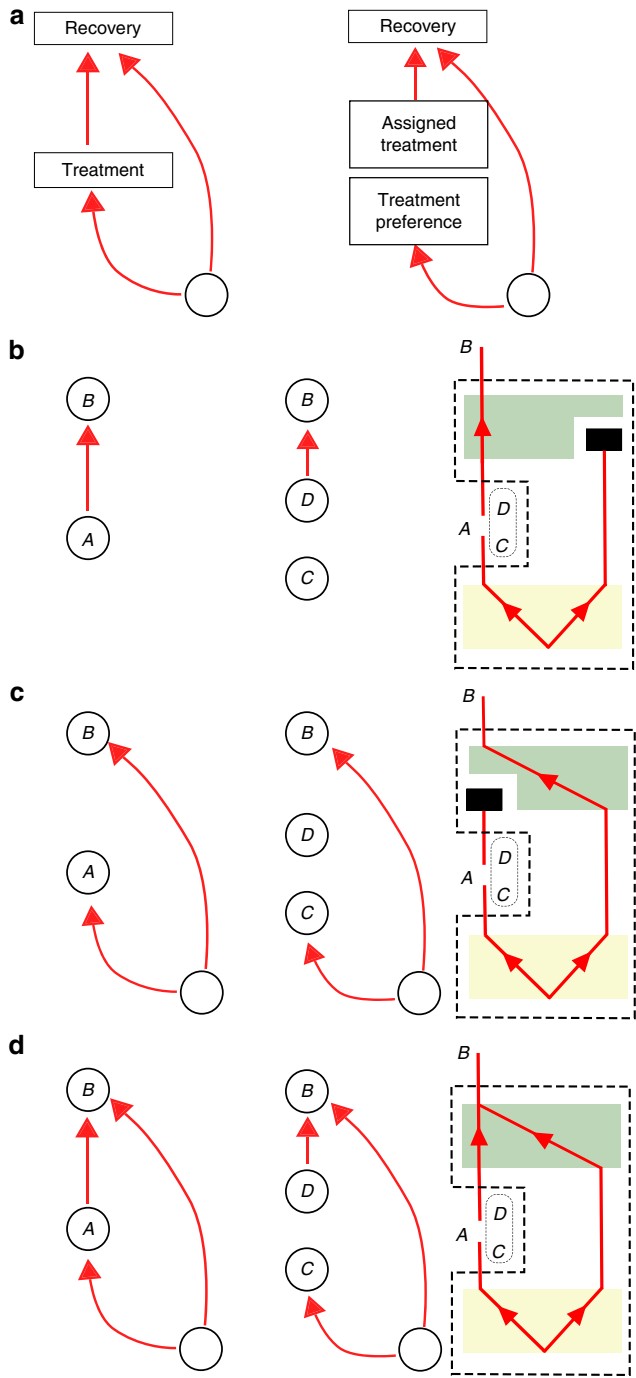

**Figure 2 | Causal relations between two time-ordered systems.**
(**a**) A drug trial aims to discern whether treatment and recovery have a cause-effect relation, whether they share an unobserved common cause, or some combination of both. To this end, pharmaceutical companies randomly assign patients to take either the drug or a placebo, so as to evaluate the cause-effect relation. One may also track treatment preference to assess the common-cause relation. A complete characterization of the causal relation requires information about both versions of the treatment variable. Abstract depictions of possible causal relations: (**b**) purely cause-effect, (**c**) purely common-cause and (**d**) general case, including mixtures of both mechanisms. In directed acyclic graphs, arrows represent influences. The variable $A$ is split into a pre-intervention version, denoted $C$, and a post-intervention version, denoted $D$. The circuits realizing the causal relations, consisting of a preparation (yellow) of $A$ and an ancilla, followed by a coupling (green) between $A$ and the ancilla, which yields $B$.

analogue of $P(CB|D) = P(CB)$), which makes it compatible with the causal structure in Fig. 2c. A causal map is said to be a probabilistic mixture of cause-effect and common-cause relations if there is a hidden classical control variable, $J$, which influences only $B$, such that for every value of $J$, either $B$ depends only on $D$ or $B$ depends only on its common cause with $C$. We show in Supplementary Note 1 that every such causal map can be expressed as having just one term of each type, $\mathcal{E}_{CB|D} = w\mathcal{E}_{B|D} \otimes \rho_C + (1-w)\rho_{CB} \otimes \mathrm{Tr}_D$, where $0 \le w \le 1$ and $\mathrm{Tr}_B\rho_{CB} = \rho_C$. The fact that the marginal on $C$ is the same in both terms follows from demanding that the control variable $J$ does not influence $C$. This demand is justified by noting that a probabilistic mixture of causal maps that are all purely cause-effect should also be purely cause-effect, but if the switch variable $J$ implementing this mixture could influence $C$ in addition to $B$, then it would itself constitute a common cause of $A$ and $B$. If a causal map is not such a probabilistic mixture, then it is termed a physical mixture of cause-effect and common-cause mechanisms.

Another distinction that is important for classifying causal relations between two quantum systems is whether the common-cause or cause-effect pathways of a given causal map are themselves quantum or not. We propose that a sufficient condition for quantumness of the common-cause pathway is that there exists an orthogonal basis of states on $D$, indexed by $d$ and denoted $\rho_d$, such that the states on $CB$ induced by these preparations, $\tau_{CB}^d \equiv \mathcal{E}_{CB|D}(\rho_d)$, are entangled. Similarly, a causal map is intrinsically quantum on the cause-effect pathway if there exists a measurement on $C$ that distinguishes a complete set of orthogonal states, indexed by $c$ and represented by projectors $\Pi_C^c$, such that the induced correlations between $D$ and $B$ are quantum for every outcome $c$. By the same reasoning established in the discussion of the quantum Berkson effect, these correlations are represented by trace-non-increasing maps from $D$ to $B$, $\mathcal{E}_{B|D}^c \equiv \mathrm{Tr}_C(\Pi_C^c\mathcal{E}_{CB|D})$, or equivalently by the normalized Choi-isomorphic states $\tau_{BD}^c$. This allows us to propose a sufficient condition for quantumness in the cause-effect pathway that closely resembles the one for the common-cause pathway: the states $\tau_{BD}^c$ must be entangled for all $c$.

These distinctions give rise to eight classes of causal maps. We here limit our attention to cases where the pathways are either both quantum or both classical, yielding four classes of interest, illustrated in Fig. 3 and termed PROBC, PHYSC, PROBQ and PHYSQ. The definition of the fifth class (COH) is the central theoretical proposal of this article: a mixture of common-cause and cause-effect relations is quantum-coherent if the causal map is intrinsically quantum in both the common-cause and cause-effect pathways and it exhibits a quantum Berkson effect. We note that the second requirement can only be satisfied if the causal map is a physical mixture, while the first implies that it is quantum in both pathways, hence COH is contained in PHYSQ. We show in Supplementary Note 2 that the inclusion is in fact strict.

**Realizing COH with a quantum circuit.** Figure 4 presents quantum circuits that realize causal relations between two qubits exemplifying each of the classes. Here $E$ denotes the system that mediates between $B$ and its common cause with $C$. System $F$ is introduced to make the gate $\mathcal{E}_{BF|DE}$ preserve dimensionality, but it is discarded afterwards. The initial state $\rho_{CE}$ in all cases is the maximally entangled state.

$$|\Phi^+\rangle \equiv \frac{1}{\sqrt{2}}(|HH\rangle + |VV\rangle), \qquad (1)$$

where $|H\rangle$, $|V\rangle$ denote the eigenstates of the Pauli operator $\sigma_z$, anticipating the identification as horizontal and vertical

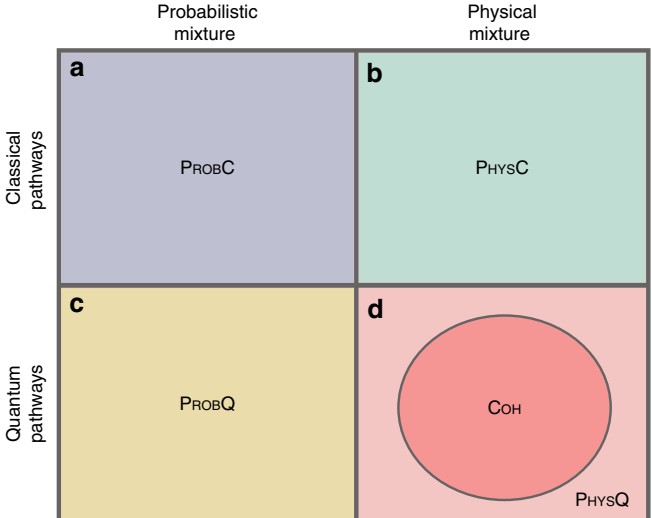

**Figure 3 | Classification of mixtures of causal relations between two quantum systems.** One can distinguish whether the common-cause and cause-effect pathways are effectively classical or whether they are quantum, and whether they are combined in a probabilistic or a physical mixture. This gives rise to four categories of interest: (**a**) a probabilistic mixture that is classical on both pathways (PROBC), (**b**) a physical mixture that is classical on both pathways (PHYSC), (**c**) a probabilistic mixture that is quantum on both pathways (PROBQ) and (**d**) a physical mixture that is quantum on both pathways (PHYSQ). We leave aside cases wherein only one pathway is quantum. The focus of this paper is the class COH, which exhibits the quantum Berkson effect and therefore describes quantum-coherent mixtures of common-cause and cause-effect relations between $A$ and $B$.

polarization states of our photonic qubits. The gate $\mathcal{E}_{BF|DE}$ that realizes a coherent mixture applies the partial swap unitary,

$$U_{BF|DE} = \tfrac{1}{\sqrt{2}}\mathbb{1}_{B|D} \otimes \mathbb{1}_{F|E} + i\tfrac{1}{\sqrt{2}}\mathbb{1}_{B|E} \otimes \mathbb{1}_{F|D}, \quad (2)$$

where $\mathbb{1}_{Y|X}$ denotes the identity operator from $X$ to $Y$. This unitary coherently combines the two-qubit identity operator, $\mathbb{1}_{B|D} \otimes \mathbb{1}_{F|E}$, which realizes a purely cause-effect relation between $A$ and $B$, and the swap operator, $\mathbb{1}_{B|E} \otimes \mathbb{1}_{F|D}$, which realizes a purely common-cause relation.

Combining the circuit elements $\rho_{CE}$ and $\mathcal{E}_{BF|DE}$ and tracing out $F$, we find

$$\mathcal{E}_{CB|D}(\cdot) = \frac{1}{2}\left[\tfrac{1}{2}\mathbb{1}_C \otimes \mathcal{I}_{B|D}(\cdot)\right] + \frac{1}{2}|\Phi^+\rangle\langle\Phi^+|_{CB}\mathrm{Tr}_D(\cdot)$$
$$-i\{\left[\tfrac{1}{2}\mathbb{1}_C \otimes \mathcal{I}_{B|D}(\cdot)\right]|\Phi^+\rangle\langle\Phi^+|_{CB}$$
$$-|\Phi^+\rangle\langle\Phi^+|_{CB}\left[\tfrac{1}{2}\mathbb{1}_C \otimes \mathcal{I}_{B|D}(\cdot)\right]\}. \quad (3)$$

The first term applies the identity channel from $D$ to $B$, $\mathcal{I}_{B|D}$, whereas the second prepares $C$ and $B$ in the maximally entangled state $|\Phi^+\rangle$. The cross terms encode coherences between these two causal relations. One can verify that this causal map is quantum in both the cause-effect and the common-cause pathway. It also exhibits a quantum Berkson effect: if $B$ is measured to be in the state $|H\rangle$, then

$$\tau_{CD}^H = \frac{1}{2}|HH\rangle\langle HH| + \frac{1}{2}|\varphi\rangle\langle\varphi| \quad (4)$$

where $|\varphi\rangle \equiv \frac{1}{\sqrt{2}}(|HV\rangle - i|VH\rangle)$, hence $\tau_{CD}^H$ is entangled. If instead $B$ is measured to be in the state $|V\rangle$, $\tau_{CD}^V$ is similarly entangled. The causal map therefore belongs to COH.

The example of PROBQ from Fig. 4c obtained by replacing the partial swap with an equal probabilistic mixture of identity and

swap, which eliminates the cross terms in equation (3). The result is a manifestly probabilistic mixture of causal relations that is nevertheless quantum on both pathways. One can further modify this gate to realize the example of PROBC in Fig. 4a complete-dephasing operations on its inputs, $D$ and $E$, which effectively reduces both qubits to classical bits.

The example of PHYSC presented in Fig. 4b also begins by complete dephasing on $D$ and $E$ to ensure that both pathways are indeed classical. The simplest example of a physical mixture would then be one, wherein $B$ is a nontrivial function of both $D$ and $E$. However, since we wish to realize all of these examples with a single experimental set-up, we consider instead a probabilistic mixture of two gates, one of which has $B$ as a nontrivial function of both $D$ and $E$, whereas the other prepares $B$ in the completely mixed state. The expression $\mathcal{E}_{CB|D}$ for each example and the proof that they are all indeed representatives of their classes are provided in the Supplementary Note 2.

**Experimental signatures of causal relations.** The four circuits of Fig. 4 are experimentally realized using the set-up of Fig. 5. The polarization degrees of freedom of different photon modes constitute the qubits in our circuit. We use downconversion to prepare the photonic modes $C$ and $E$ in the maximally entangled polarization state $|\Phi^+\rangle$.

To realize our example of COH (Fig. 4d), the partial swap in equation (2) is implemented using linear optics[23]. Here we significantly improve the stability of the experimental concept of ref. 23 by incorporating the working principle around a displaced Sagnac interferometer. The other three examples from Fig. 4 are obtained by variations on this set-up. Delaying the photon in mode $E$ relative to the one in mode $D$ prevents two-photon interference at the first beam splitter of the Sagnac interferometer, so that the interferometer implements a probabilistic mixture of identity and swap operations (our example of PROBQ, Fig. 4c). This latter circuit realizes the same causal map implemented in ref. 3, which focused on the task of resolving probabilistic mixtures of cause-effect and common-cause relations. However, the experimental set-up of ref. 3 could not realize physical mixtures, which are required to address the broader question, investigated in the present work, of how these two extremes may be combined in general.

Both causal pathways can be made classical by passing the modes through completely dephasing channels on $D$, $E$ and $B$. The example of PROBC from Fig. 4a is realized by dephasing in the $\{|H\rangle, |V\rangle\}$ basis on all three. The example of PHYSC from Fig. 4b is also achieved by implementing complete dephasing, but in different bases: $\{|R\rangle, |L\rangle\}$ on $D$, $\{|D\rangle, |A\rangle\}$ on $E$ and $\{|H\rangle, |V\rangle\}$ on $B$. For further details on how to implement the four example classes of causal structures using a single experimental set-up, see Supplementary Note 3.

We characterize the causal maps realized in the experiment using tomography[9,12,16,24]: measurements on $C$ and $B$ and preparations on $D$, each ranging over the six eigenstates of Pauli observables, allow us to reconstruct the map using a least-squares fit. The causal maps obtained from the four circuits in Fig. 4 are shown in Fig. 6 and achieve fidelities above 93% with their respective targets. Although these maps encode a complete description of the causal relation realized between $A$ and $B$, since our goal is only to classify the causal relation, we will introduce and evaluate specific indicators that can achieve this purpose with fewer measurements and preparations.

A witness of physical mixture (as opposed to probabilistic) can be evaluated using only measurements of the Pauli observables $\sigma_x$ on $C$ and $\sigma_z$ on $B$, with outcomes $c, b = \pm 1$, while preparing the $d$ eigenstate of $\sigma_y$ on $D$, with $P(d = \pm 1) = \frac{1}{2}$. (Different choices of Pauli observables generate a family of such witnesses.) For subsets

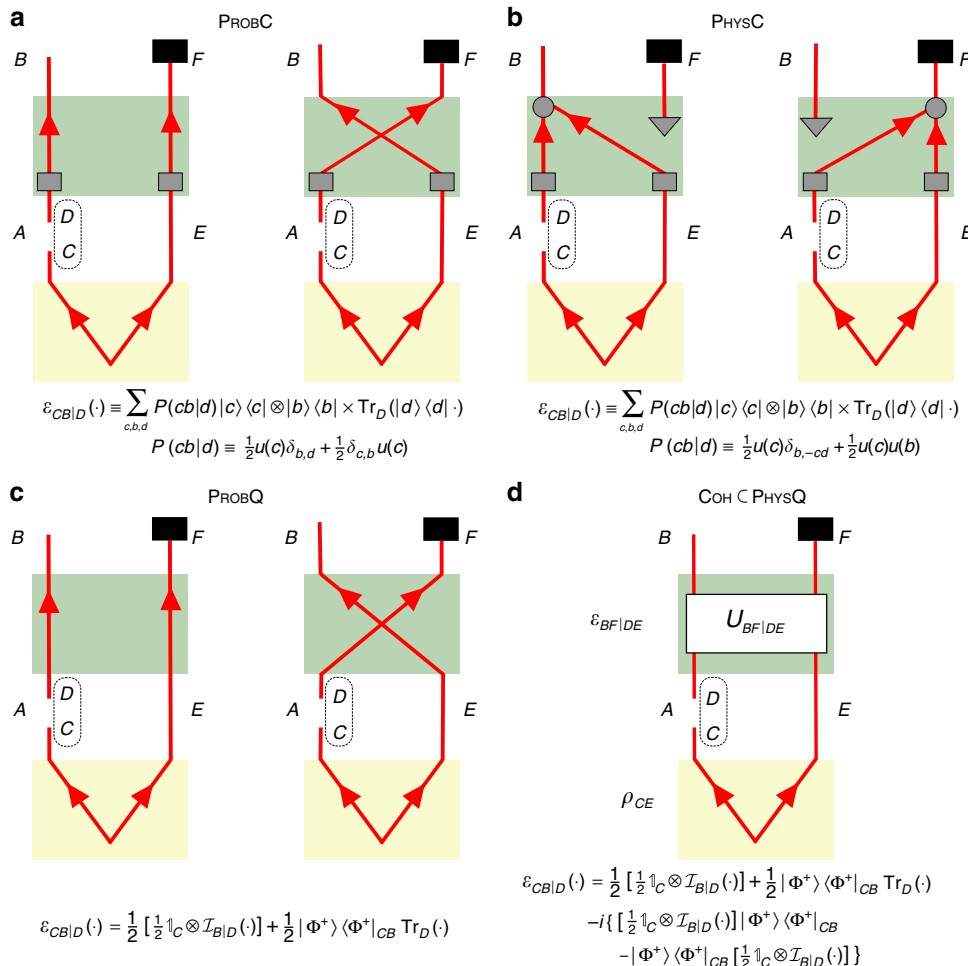

**Figure 4 | Quantum circuits which realize examples of different classes of causal relations between two qubits.** Circuits for four combinations of cause-effect and common-cause mechanisms: (**a**) PROBC, (**b**) PHYSC, (**c**) PROBQ and (**d**) COH ⊂ PHYSQ. In all circuits, $C$ and $E$ are initially prepared in the maximally entangled state $|\Phi^+\rangle$ and $F$ is discarded at the end. The examples differ only in the choice of the gate $\mathcal{E}_{BF|DE}$, as described in the text. Panels with two circuits represent an equal probabilistic mixture of both scenarios. Black squares represent the trace operation, grey squares represent complete-dephasing operations, grey triangles represent preparations of the completely mixed state, grey circles represent the classical XNOR gate, which sets $b = -ed$ for binary variables $b,e,d$ taking the values $\{\pm 1\}$ and the two-qubit unitary $U_{BF|DE}$ is the partial swap given by equation (2). Below each example, we specify the causal map $\mathcal{E}_{CB|D}$ obtained from the state $\rho_{CE}$ and the gate $\mathcal{E}_{BF|DE}$ via $\mathcal{E}_{CB|D}(\cdot) = \mathrm{Tr}_F \circ \mathcal{E}_{BF|DE}(\cdot \otimes \rho_{CE})$. Lowercase letters $c,b,d$ represent classical binary variables, $P(cb|d)$ represents a conditional probability distribution over these, $\delta_{x,y}$ denotes the Kronecker delta function and $u(x)$ denotes the uniform distribution over $x$.

of this data with different values of $b$, one can compute the covariance of $c$ and $d$, which we denote $\mathrm{cov}(c,d|b)$ (see Supplementary Note 5 for details). Letting $P(b)$ denote the probability of obtaining the outcome $b$, we define our witness to be

$$\mathcal{C}_{CD} \equiv 2\sum_b bP(b)^2\mathrm{cov}(cd|b). \qquad (5)$$

We show in Supplementary Note 5 that $\mathcal{C}_{CD}=0$ for all probabilistic mixtures of common-cause and cause-effect, which implies that $\mathcal{C}_{CD} \neq 0$ heralds a physical mixture.

We reconstruct the operators $\tau_{BD}^c$, $\tau_{CB}^d$ and $\tau_{CD}^b$ using subsets of tomographic data (for example, using only runs that found $B$ in the state $|H\rangle$ to reconstruct $\tau_{CD}^H$). The entanglement of these states is quantified by the negativity[25],

$$\mathcal{N}_{XY}^z \equiv \frac{1}{2}\left(\mathrm{Tr}\left|T_Y\left(\tau_{XY}^z\right)\right| - 1\right), \qquad (6)$$

where $T_Y(\cdot)$ denotes transposition on $Y$. Quantumness in the cause-effect and common-cause pathways is therefore witnessed

by $\mathcal{N}_{BD}^c > 0 \,\forall c$ and $\mathcal{N}_{CB}^d > 0 \,\forall d$, respectively, while $\mathcal{N}_{CD}^b > 0 \,\forall b$ witnesses a quantum Berkson effect. See Supplementary Note 4 for more details on obtaining the negativity of the pre- and post-selected states from experimental data.

Figure 7 summarizes the values of these witnesses for the four circuits of Fig. 4 along with the corresponding theoretical expectations. The indicators $\mathcal{N}_{BD}^c$, $\mathcal{N}_{CB}^d$ and $\mathcal{N}_{CD}^b$ are evaluated using tomographic reconstructions of the operators $\tau_{BD}^c$, $\tau_{CB}^d$ and $\tau_{CD}^b$, respectively, under preparations (on $D$) and measurements (on $C$ and $B$) of $\{|H\rangle, |V\rangle\}$. Scenarios (b) and (d) show evidence of a physical mixture, with $\mathcal{C}_{CD}=0.40\pm0.02$ and $\mathcal{C}_{CD}=0.46\pm0.02$, while scenarios (c) and (d) exhibit quantumness in the common-cause and cause-effect pathways. We find $\mathcal{N}_{CB}^d$ and $\mathcal{N}_{BD}^c$ non-zero for $c, d \in \{H,V\}$. These signatures confirm that we realized physical mixtures and quantum common-cause and cause-effect mechanisms as intended.

The most important indicator for our purposes is $\mathcal{N}_{CD}^b$, which verifies the quantum Berkson effect. As expected, scenarios (a–c) have $\mathcal{N}_{CD}^b=0$ (to within statistical error) and are therefore compatible with an incoherent mixture of common-

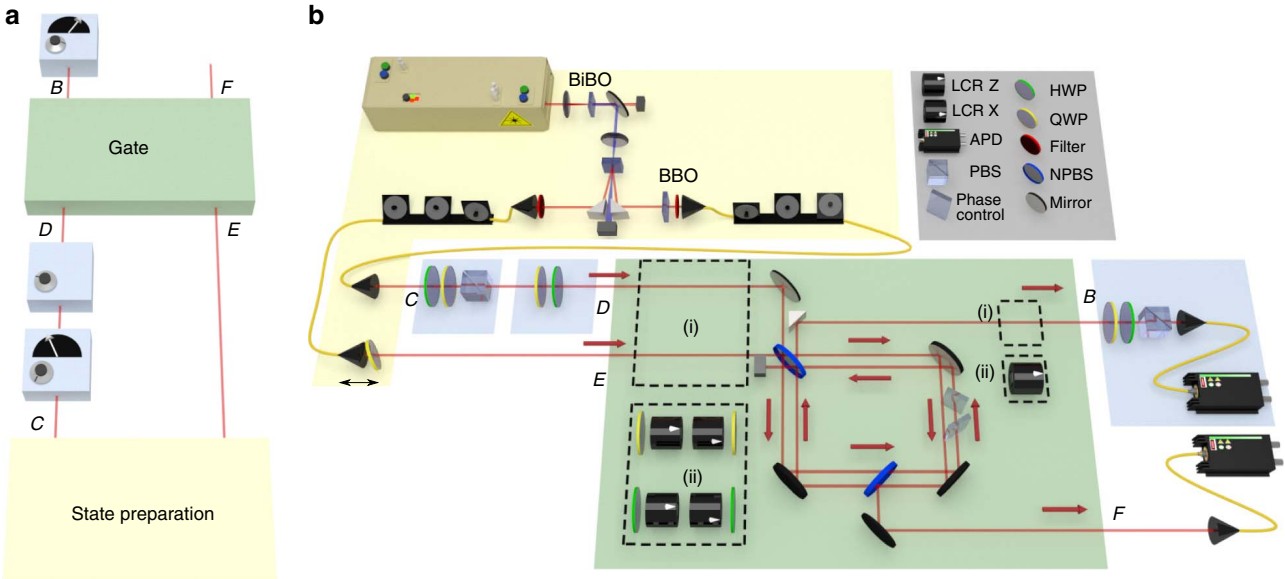

**Figure 5 | Optical implementation of different causal relations.** (**a**) Schematic diagram of the experiment. The gate allows the implementation of different ways of combining a common-cause relation (between *C* and *B*) with a cause-effect relation (between *D* and *B*). The initial preparation targets the maximally entangled state $|\Phi^+\rangle$. One photon is measured at *C* and reprepared at *D*; then both are sent through the gate. The photon at *B* is detected in coincidence with the photon at *F* to post select only on those cases wherein a pair was produced. (**b**) Experimental set-up, including the polarization-entangled photon source and the partial swap, which implements the unitary $U_{BF|DE} = \frac{1}{\sqrt{2}}\mathbb{1}_{B|D} \otimes \mathbb{1}_{F|E} + i\frac{1}{\sqrt{2}}\mathbb{1}_{B|E} \otimes \mathbb{1}_{F|D}$ by tilting the glass plates to a specific angle in the Sagnac interferometer. (i) For the quantum mixtures, no dephasing is applied and a translation stage adjusts the delay of the photon at *E* with respect to the one at *D*. (ii) For the classical mixtures, the LCRs and wave plates are used to apply complete dephasing on *D*, *E* and *B*, respectively. Notation for optical elements: bismuth-borate (BiBO), β-barium-borate (BBO), half-wave plate (HWP), quarter-wave plate (QWP), liquid-crystal retarder (LCR), polarizing beam splitter (PBS), non-polarizing beam splitter (NPBS), avalanche photo diode (APD).

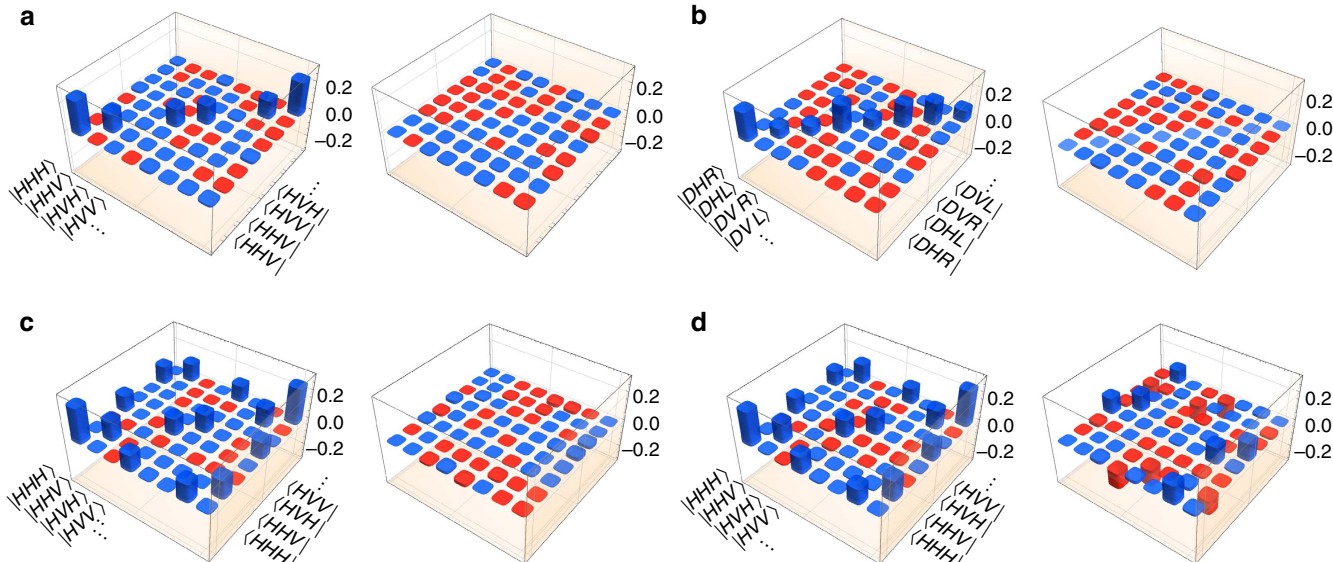

**Figure 6 | Reconstructed causal maps.** Reconstruction of the real and imaginary parts of the Choi state $\tau_{CBD} = \mathrm{Tr}_{FD'}\left[ \left( \mathcal{E}_{BF|D'E} \otimes \mathcal{E}_{CD} \right) \left( |\Phi^+\rangle\langle\Phi^+|_{D'D} \otimes \rho_{CE} \right) \right]$ when targeting four combinations of cause-effect and common-cause mechanisms: (**a**) PROBC, (**b**) PHYSC, (**d**) PROBQ and (**d**) COH. The causal maps for **a,b** are given in the local bases that diagonalize the target map, to make explicit their classical nature; no such bases exist for **c,d**. Blue (red) colour bars represent positive (negative) values. The fidelities[30], $F \equiv \left[ \mathrm{Tr}\sqrt{\tau^{\frac{1}{2}}\tau_{\mathrm{th}}\tau^{\frac{1}{2}}} \right]^2$, to the theoretically calculated Choi state $\tau_{\mathrm{th}}$ are high, at $(98.1 \pm 0.2)$%, $(98.06 \pm 0.08)$%, $(97.1 \pm 0.1)$% and $(93.7 \pm 0.3)$%, respectively, verifying that the experiment is performing as intended. Uncertainties on fidelities indicate one s.d. and are estimated using Monte-Carlo simulations with Poissonian noise on photon counts. Notation for polarization states: $|H\rangle$, horizontal, $|V\rangle$ vertical, $|D\rangle = 1/\sqrt{2}(|H\rangle + |V\rangle)$ diagonal, $|A\rangle = 1/\sqrt{2}(|H\rangle - |V\rangle)$ anti-diagonal, $|R\rangle = 1/\sqrt{2}(|H\rangle + i|V\rangle)$ right-circular, $|L\rangle = 1/\sqrt{2}(|H\rangle - i|V\rangle)$ left-circular.

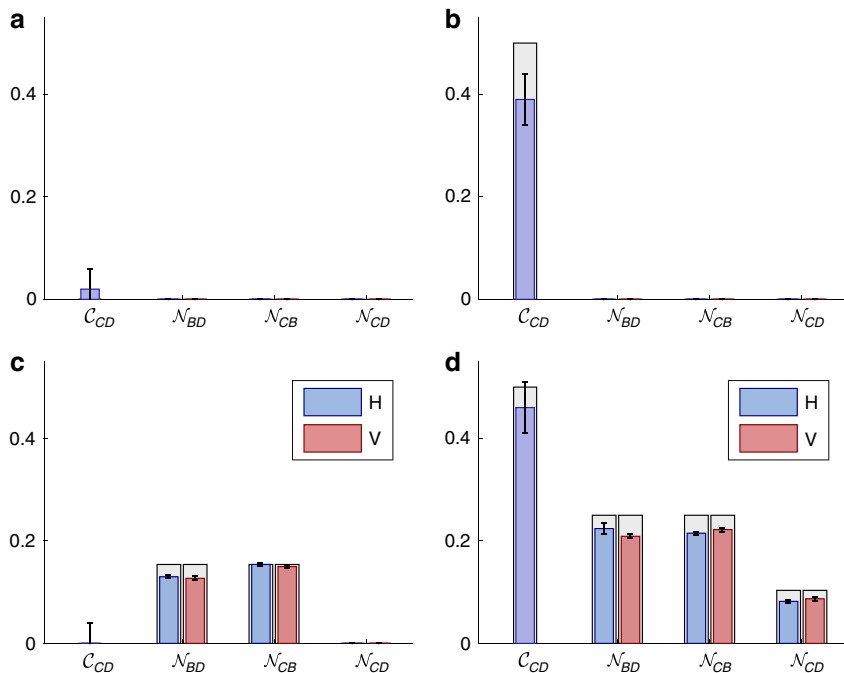

**Figure 7 | Classifying causal relations using induced correlations.** For each circuit in Fig. 4, we present theoretical (grey) and experimental (coloured) values for the different witnesses of causal relations. Circuits **b,d** have $\mathcal{C}_{CD} \neq 0$, witnessing physical mixtures, whereas **a,c** are consistent with probabilistic mixtures, since $\mathcal{C}_{CD}$ is zero within 1 s.d. Circuits **c,d** show evidence of intrinsically quantum cause-effect and common-cause mechanisms, $\mathcal{N}_{BD}^{c} \neq 0$ and $\mathcal{N}_{CB}^{d} \neq 0$ for d, c = H, V, whereas **a,b** are consistent with classical mechanisms. Only **d** has $\mathcal{N}_{CD}^{b} \neq 0$ for b = H, V, witnessing a quantum Berkson effect. Uncertainties indicate 1 s.d. and are estimated using Monte-Carlo simulations, assuming Poissonian noise on the photon counts.

cause and cause-effect relations. Scenario (d), however, exhibits Berkson-type-induced entanglement, with $\mathcal{N}_{CD}^{H} = (0.083 \pm 0.003)$ and $\mathcal{N}_{CD}^{V} = (0.087 \pm 0.004)$ This, combined with the evidence of quantumness of each individual mechanism, constitutes a clear signature of the class COH, a quantum-coherent mixture of cause-effect and common-cause relations.

## Discussion

*A priori*, it is not obvious how one ought to define a quantum-coherent combination of causal relations, in particular a quantum-coherent combination of common-cause and cause-effect relations. In this article, we have proposed a particular definition and demonstrated the possibility of realizing such quantum coherence experimentally. There are two distinct notions of quantum coherence that one might think are pertinent to our problem, and both feature in our definition.

The first notion of quantum coherence applies to elements of a set of alternatives that are jointly exhaustive and mutually exclusive, such as the eigenstates of some observable. In this case, an incoherent mixture is a probabilistic mixture of the alternatives and consequently it is reasonable to define a state as exhibiting coherence whenever it cannot be expressed as such a probabilistic mixture. However, the sorts of causal relations that we here seek to combine coherently, a cause-effect relation and a common-cause relation, do not constitute mutually exclusive alternatives. A pair of systems may be connected by both a cause-effect relation and a common cause. The possibility of two causal mechanisms acting simultaneously necessitates the category of physical mixtures of causal mechanisms. Given that such physical mixtures can arise classically, the mere inapplicability of a probabilistic mixture should not lead one to infer the presence of quantumness. This is why we use additional criteria for judging a combination of causal relations to be a

quantum-coherent combination. Because physical mixtures are distinguished by the strength of the induced correlations in the Berkson effect, we have proposed that a necessary condition for having a quantum-coherent mixture is that the Berkson-induced correlations exhibit entanglement.

The second notion of quantum coherence is the one relative to which different systems are said to be coherent with one another: for independent systems, this occurs when their joint state is entangled, for the input and output of a quantum channel, this occurs when the channel is not entanglement breaking. In the causal context, therefore, a common-cause relation between a pair of systems can be judged coherent if the state of the systems is entangled, while a cause-effect relation between a pair of systems can be judged coherent if the associated channel is not entanglement breaking. This second notion of coherence is applicable, therefore, to individual causal pathways rather than the manner in which they are combined. Consequently, we have proposed that another necessary condition for a mixture of cause-effect and common-cause relations to be quantum-coherent is that each of the pathways, common-cause and cause-effect, are themselves coherent.

Our approach to defining quantum-coherent combinations of different causal relations differs significantly from the one suggested in recent work seeking to define superpositions of different causal orders. The proposal for witnessing causal nonseparability in ref. 26, for instance, judges the causal order between a pair of systems to be quantum indefinite whenever the causal map cannot be written as a probabilistic mixture of terms with definite causal orders. However, from our perspective, *A* causing *B* is not necessarily mutually exclusive to *B* causing *A* (just as a common-cause relation is not mutually exclusive to a cause-effect relation). To imagine both acting simultaneously— which we would term a physical mixture of the two cause-effect relations—is simply to imagine the possibility of causal cycles.

This is an exotic possibility, but one that is classically meaningful. As such, having a causal map that is not a probabilistic mixture of causal orders need not, by itself, be evidence of quantumness. See Supplementary Discussion 1 for more on the related topic of superposition of causal orders.

As we progress to studying more complex scenarios, for instance, involving a larger number of systems, we are likely to find an even wider range of types of coherence, resembling the many types of entanglement that arise when more than two parties are involved. The problem of classifying the causal possibilities in this case—in particular quantum-coherent mixtures of various relations—is significantly more complex than the one considered here. Developing such a classification for an arbitrary number of systems with arbitrary dimensionality constitutes an important pillar in the new research programme that seeks to understand causality in quantum theory.

The understanding of uniquely quantum, combinations of common-cause and cause-effect relations introduced in this article also has implications for the topic of non-Markovianity. In the context of the dynamics of open quantum systems, the assumption of Markovianity states that the environment with which the principal system interacts has no memory and hence is unable to preserve a record of earlier states of the system. (See ref. 27,28 for a review of proposed quantum statements of Markovianity.) This assumption simplifies the mathematical treatment of the system's dynamics considerably, but in most realistic models it holds only approximately, which has sparked considerable interest in quantum non-Markovianity in recent years. From the perspective of causal modelling, non-Markovianity arises when the environment acts as a common cause of the system at different times (in addition to the cause-effect relations arising from the evolution of the system itself). As such, the fact that there are intrinsically quantum ways of mixing common-cause and cause-effect relations implies a greater variety of types of non-Markovianity than one sees classically.

## Methods

**Photon source.** We produce polarization-entangled photon pairs using spontaneous parametric downconversion in two type-I nonlinear crystals. We begin with a Ti:Sapphire laser, centred at 790 nm with a spectral bandwidth of 10.5 nm, a repetition rate of 80 MHz and an average power of 2.65 W. The laser light is frequency doubled in a 2-mm thick bismuth-borate crystal, which creates a pump beam of 0.65 W centred at 395 nm with a 1 nm full-width at half-maximum bandwidth. With two cylindrical lenses, the pump is focused onto a pair of 1 mm $\beta$-barium-borate crystals with orthogonal orientations for type-I spontaneous parametric downconversion. Bandpass filters are placed to reduce background noise from the pump. Additional compensation crystals are used to counteract the effects of temporal and spatial walkoff[29]. Polarization-entangled photon pairs at 790 nm are prepared in the state $|\Phi^+\rangle$, which we achieve with $(96.31 \pm 0.08)\%$ fidelity. Inteference filters on both sides set the photon bandwidths to 3 nm. The photons are then coupled into single mode fibres and sent towards the partial swap. The polarization is set with polarization controllers and the phase of the entangled state is tuned by tilting a quarter-wave plate (QWP) at the output of one of the fibres.

**Implementing the partial swap.** The partial swap uses a folded displaced Sagnac interferometer, with two 50/50 beam splitters and two NBK-7 glass windows, which are counter rotated to set the phase with minimal beam deflection. The visibility of the Sagnac interferometer without background subtraction is $(93.6 \pm 0.1)\%$. This is measured by blocking one input ($D$ or $E$) to the gate and measuring the number of photons at the output $B$ as a function of the window angles in the Sagnac interferometer. For the coherent partial swap to be effective, photon pairs in modes $D$ and $E$ must undergo two-photon quantum interference on a beam splitter before entering the gate. Hong–Ou–Mandel (HOM) interference between photons input at $D$ and $E$ is measured at the first beam splitter using a translation stage on input $E$. A dip in visibility of $(95 \pm 2)\%$ is achieved. To implement the gate for the class ProbQ, a delay of 3 ps is added to photon $E$, which removes the interference. For all other cases, the delay is set to a value corresponding to the centre of the HOM dip to maximize the two-photon interference.

**Dephasing channels.** The dephasing channels before and after the Sagnac interferometer are implemented using variable liquid-crystal retarders (LCR), which exhibit a voltage-dependent birefringence, introducing a relative phase of 0 or $\pi$ on orthogonal polarization states. Probabilistic dephasing is achieved by switching them on and off at random at a rate of 10 Hz, with probability 1/2 of implementing a $\pi$ phase shift during each interval. The input and outputs can be dephased on a different polarization bases. Dephasing along the $\{|D\rangle, |A\rangle\}$ basis is achieved with the LCR axis at 0°, and in the $\{|H\rangle, |V\rangle\}$ basis at 45°. Two QWPs on either side of the LCR after $D$ and two half-wave plates (HWP) on either side of the LCR after $E$ are used to rotate between the different dephasing bases required for the classical mixtures.

**Measurement procedure.** The experiment proceeds in the following way. The unitary $U_{BF|DE}$ is set by adjusting the window angles in the Sagnac interferometer such that the phase difference between the two paths is $\pi/2$. A HOM dip is then measured and the arrival time of the photons is set with a translation stage at $E$. The entangled state on $C$ and $E$ is initialized by preparing and measuring remote entanglement between $C$ and $B$. The polarization is measured using a HWP, QWP and polarizing beam splitter (PBS) in sequence. The HWP and QWP are adjusted so that one polarization state can pass through the PBS. The polarization at $C$ is measured with a HWP and QWP, and assuming the photon has passed through the PBS, the polarization is reprepared with another QWP and HWP at $D$. Photons are then sent to the partial swap gate, and the polarization at $B$ is measured using another QWP and HWP. Coincidence counts at $F$ and $B$ are measured using silicon avalanche photodiodes and a coincidence logic with a coincidence window of 3 ns. Coincidences are measured at a rate of ∼1 kHz. We measure the different combinations of polarization eigenstates and repeat the procedure for the four different causal scenarios.

**Code availability.** The code used to generate the findings of this study are available from the corresponding author on reasonable request.

**Data availability.** The data that support the findings of this study are available from the corresponding author on reasonable request.

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

## Acknowledgements

This research was supported in part by the Foundational Questions Institute (Grant Number FQXI-RFP-1516), the Natural Sciences and Engineering Research Council of Canada (NSERC), Canada Research Chairs, Industry Canada and the Canada Foundation for Innovation (CFI). Research at Perimeter Institute is supported by the Government of Canada through Industry Canada and by the Province of Ontario through the Ministry of Research and Innovation.

## Author contributions

R.W.S., K.R. and K.J.R. conceived the original idea for the project. K.R. and R.W.S. developed the project and the theory. J.-P.W.M. and K.J.R. designed the experiment. J.-P.W.M. performed the experiment and the numerical calculations. J.-P.W.M., K.R., R.W.S. and K.J.R. analysed the results. J.-P.W.M. and K.R. wrote the first draft of the paper and all authors contributed to the final version.

## Additional information

**Competing interests:** The authors declare no competing financial interests.

**Publisher's note**: 

