## [Peer Review File · Nature Communications]

Reviewer #1 (Remarks to the Author):

Dear editor

In this letter the authors present a work exploring causal relations in a quantum bipartite system. I think that this work is interesting, but rather technical, not very accessible to a generic audience and of interest for specific readers.

Also the presented experiment is of interest and sound, but rather standard.

For these reasons, I suggest publishing it in a more specific journal.

Reviewer #2 (Remarks to the Author):

The manuscript deals with various kinds of causal relations that interpolate between common cause and cause-effect.

The authors distinguish classical and "quantum" causal relations, introducing notions of "quantum common cause" (two quantum systems in an entangled state) and "quantum-cause-effect relation" (one quantum system obtained from another through a coherent quantum transformation).

Moreover, the authors provide criteria to distinguish different types of causal relationships and illustrate the distinction in a photonic experiment.

The general topic is interesting and the manuscript is nicely written. However, my opinion is that the degree of innovation is not sufficient for publication in Nature Communications. Characterizing quantum states and quantum maps in relation to entanglement is a standard approach and a large part of the manuscript is just a translation/application of existing notions to the topic of causal relations.

While such translation can be interesting, the authors seem to have gone a bit too far when using new names like "causal maps" and "causal tomography" for already well established notions. The theory of witnesses is also well understood and its application to the causal scenario is relatively simple. The properties of the partial swap are well-known in quantum computing and the experimental implementation of this two-qubit gate has become standard by now.

Another aspect that I find unsatisfactory is the motivation. The authors motivate the study of quantum causality in general, but only provide vague motivations for the specific work done in this manuscript. Quantum gravity, foundational puzzles, and quantum technologies are briefly mentioned, but without any direct connection with the results.

Finally, two points in the manuscript are relatively weak. First, basing the definition of quantum causal relations on entanglement is one possibility, but not the only choice. "Entanglement" is just a specific instance of "quantumness" or "coherence", although the authors use these terms interchangeably.

Second, the authors define the "quantum Berkson effect" as the conditional generation of entangled states through post selection. But Berkson's paradox is not only the observation that post selection can induce correlations between two uncorrelated variables. Taking this straightforward observation and replacing classical with quantum correlations (however defined) is fine, but it is not enough to make an interesting quantum version of Berkson's paradox.

Reviewer #3 (Remarks to the Author):

The paper investigates the possible causal relations that can exist between two time-ordered quantum systems. The authors distinguish 4 classes of causal relations, whether the systems are linked by classical or quantum pathways, and by probabilistic or "physical" mixtures of common-cause and cause-effect relations. Inside the class 'PHYSQ' (physical mixture with quantum pathways), the additional subclass 'COH' is identified, where the systems exhibit a quantum

version of the Berkson paradox. The authors construct witnesses to test which class the causal relations between a pair of systems belongs to. An experiment is carried out, where variations of the same setup allow the authors to demonstrate all different classes of quantum causal relations they identified.

The study of quantum causal relations has gained a lot of interest lately in the field of quantum foundations. The in-depth analysis of the various causal relations presented in the paper is thus particularly timely and interesting. The paper contains a wealth of information, is clearly written and well-organised. The data analysis in the experiment is well-described and appropriate.

There is just one aspect in the derivation of a witness for "physical mixtures" which did not convince me, as I explain below. Before I can recommend the paper for publication, I would need the authors to address that point.

I also make below a few remarks and suggestions that the authors may want to consider to further improve the clarity of the paper, and list a few typos in some equations of the Supplementary Information that I spotted.

*** Witness of physical mixture ***

After Eq.(S65) of the Supplementary Information, and in the following derivation, it seems to be implicitly assumed that $\text{tr}_B \rho_{CB}$ from the first term of (S65) (corresponding to the common-cause case) and ρ_C from the second term (corresponding to the cause-effect case) are the same. I do not see why this should necessarily be the case: couldn't we consider a probabilistic mixture of common-cause and cause-effect relations, for which the restrictions to the system C do not coincide?

Consider for instance the following 2 (classical) causal maps:

$$P_{ce}(cbd) = P_{ce}(c) P_{ce}(b|d) u(d) = \frac{1}{2} \delta_{c,+1} \delta_{b,d} \frac{1}{2},$$

$$P_{cc}(cbd) = P_{cc}(c) P_{cc}(b|c) u(d) = \frac{1}{2} \delta_{c,-1} P_{cc}(b) \frac{1}{2}.$$

Clearly, P_{ce} and P_{cc} are of the cause-effect and common-cause forms, respectively.

Define now the probabilistic mixture $P = \frac{1}{2} * P_{ce} + \frac{1}{2} * P_{cc}$. Calculating the witness C_{CD} of Eq.(S77), I find $C_{CD} = \frac{1}{2}$ (the same value as theoretically predicted for the experimental implementation of PHYSC and COH), in contradiction with the claim that $C_{CD} = 0$ for all probabilistic mixtures of cause-effect and common-cause pathways.

It thus seems to me that the witness proposed by the authors to identify physical mixtures of cause-effect and common-cause pathways is not valid in general, without the additional unjustified (?) assumption that " ρ_C " is the same for both pathways. Another witness would need to be constructed and measured to make the experiment conclusive in this regard.

(Another argument can indeed be given to show that the observed correlations in the PHYSC and PHYSQ experiment are physical mixtures.)

Or did I misunderstand something?

*** Other remarks / questions ***

Main text:

- On the choice of terminology: to me, what the authors call a "coherent mixture" would be a quantum superposition; to refer to what they call a "physical mixture" of cause-effect and direct cause, I would have talked about a "combination" of cause-effect and direct cause.

I was wondering if there was a specific reason, why the authors have chosen the same

terminology of "mixture" for all situations? Some comment may be useful.

(It seems to me that "superposition" is more common in the literature, while it's not used at all in the paper. Is "physical mixture" a standard terminology, or one that is introduced by the authors?)

- p.3, l.51, "realizing this possibility requires exotic new physics": note that the "quantum switch" has been realised in a rather standard photonic circuit.

- p.9, l.122, why is it required that τ_{CD}^b is entangled for all b ? Why would it not be enough that there just exists one b , for which it is entangled? Is there a counter example? Some comment here would be useful.

Same questions for τ_{CB}^d and τ_{BD}^c on p.10.

- p.9, l.131-132: the result is proven in the Supplementary Material for the classical case; some argument (or at least some quick comment) should be given to justify that it also applies to the quantum case.

- p.10, l.137, why is it a sufficient condition, and not a necessary one? Again, some clarification would be useful.

- The definition of the class COH requires that (i) the causal map is quantum in both pathways and (ii) it exhibits a Quantum Berkson Effect(QBE).

This suggests that one may have a QBE outside of the class PHYSQ? Where else would that be possible, and what would it look like?

- In the classification of the different classes of mixtures: I guess we could have situations where only one pathway (cause-effect or common cause) is quantum, the other one being classical? That would be situations that fit in none of the various classes. Would the authors like to comment on that?

- Figure 4: I find it rather confusing that the red lines connect C and D. I would suggest to draw them as in Fig. 2.

In the caption, 3rd line, by "equal mixture" I guess one means "equal probabilistic mixture"? 4th line, one may clarify in which basis the "complete dephasing operations" are performed (or say that different bases are considered).

- p.19, l.240: it may be misunderstood that only preparations and measurements in the $\{H,V\}$ basis are required to evaluate the negativities. One may clarify here that a full topography is actually needed (as described in the Suppl. Info.).

Supplementary Information:

- p.1, is E really needed to mediate the influence of λ ? Couldn't one simplify the analysis and let λ directly influence B ?

In "Suppose that the conditional $P(B|DEJ)$ reduces to $P(B|DJ)$ for all values of J ", I guess that should also be for all values of D ?

- Just before Eq. (S8), it may be useful to recall that ρ_{CE} will always be taken to be Φ^+ .

- In the paragraph that follows Eq. (S17), it may be useful to refer to Eq. (S12) to clearly see which "two-qubit identity operator" and "swap operator" one is referring to.

- In Eq.(S32), $|c\rangle\langle c|$ is not defined. It may be useful also to say how it is linked to $|e\rangle\langle e|$, and clarify how $P(cb|d)$ is obtained from $P(bf|de)$.

- It seems that Eq. (S33) assumes that $\{|d\rangle\}$ is the basis of the Choi-isomorphism? But this is not the case in the analysis that follows, where the Y basis will be used (cf e.g. Eq. (S53)). This should be clarified.

- It may be useful also to clarify that Eq.(S34) is obtained for $\rho_d = |d\rangle\langle d|$ and $\Phi_C^c = |c\rangle\langle c|$.

- Sometimes the indices c,b,d are taken to be 0,1; sometimes they are taken to be ± 1 . It may be useful to clarify this, and when the authors switch from one to the other.

- Top of p.6, 2nd line, I understand that the result is proven only if c and d have uniform prior distributions?

- Bottom left of p.6, in "However, the causal map cannot be realized by a probabilistic mixture of purely common-cause and purely cause-effect relations, since the last term sets $b = c \oplus d$ ". Is that argument really enough? Couldn't one imagine a more complicated probabilistic mixture of common-cause and cause-effect relations, where the two terms combine in such a way that the relation $b = c \oplus d$ appears in the combination, while it would be neither in one term, nor in the other?

I feel that some stronger argument should be given here, to explain why this cannot be the case.

- Bottom left of p.7, it may be useful to say that the map $\{\text{cal } E\}_{\{BF|DE\}}$ is modified from the original (S45), not from (S48).

*** A few typos in the Supplementary Information ***

- Eqs. (S15-S21), there are some $1/\sqrt{2}$ and $1/2$ factors missing (where exactly they are missing depends on how the authors want to define A_k and B_k).

- Eq. (S25), I believe there is a d_B factor missing (as in (S7)). It then sounds a bit weird to read, just before, that " $[\tau_{CB}^d]$ does not require renormalization to ensure unit trace".

- There is a $1/2$ factor missing in Eq. (S29) (which should be consistent with (S22)).

- I believe there is a $1/2$ factor missing in Eq. (S50).

- Eq. (S53), I found the same 4 terms, but with all $\pm y$ states flipped (so that $b \oplus c \oplus d = 0$, rather than 1).

Am I correct also to think that $P(cb|d)$ in that equation is not the same as in the definition of the maps (e.g. from Fig.4), but differs by a flip of d (due to its measurement in the Y basis, and the partial transpose being applied somewhere...)?

- Eq. (S75), there should be a partial transpose on system D. (And putting it last may look more natural?)

Reviewer #4 (Remarks to the Author):

The authors theoretically describe and experimentally demonstrate various classes of causal

structures that can occur in a two-qubit quantum system. This is an interesting and timely result in the novel field of quantum causality, which has led to important recent developments in quantum foundations.

Although in parts quite dense, the manuscript is well written and sufficiently introduces the studied concepts and the experimental design. The Supplementary Information is very comprehensive on both the experimental and theoretical aspects of the work and complements the paper. However, I feel that the excellent technical presentation comes somewhat at the cost of the discussion of the physical implications and related work. I have made a few specific suggestions below.

Overall the quality of the manuscript is very high and upon extension of the discussion, I can recommend publication in Nature Communications.

- 1) In the second paragraph on page 3 the authors mention that the present work is based on Ref.[3,13]. Of these Ref.[3], which shares many of the present authors, uses a similar experimental setup to study cause-effect and common-cause structures in quantum mechanics. I would recommend to expand on the relation between the current work and Ref.[3].
- 2) The authors show in the supplementary information that the example from the class PHYSC can be realized by implementing dephasing in different bases. What is the physical intuition behind the choice/requirement for different bases?
- 3) It would be interesting to discuss the power of the introduced witness of physical mixture. The authors show that it is zero for all probabilistic mixtures, but it is not discussed what fraction of physical mixtures it can detect.
- 4) There seems to be a typo in the caption of Fig.6, where left-circular is denoted $|R\rangle$ and right-circular $|L\rangle$, respectively.
- 5) The discussion section focuses mainly on the case of multiple mechanisms, but is missing a discussion of the physical implications of coherent mixtures of causal mechanisms. Specifically, it would be interesting to discuss what the discovery of such situations contributes to our understanding of quantum causality or any potential applications as in the case of a superposition of causal orders.

Reviewer #1:

Dear editor

In this letter the authors present a work exploring causal relations in a quantum bipartite system. I think that this work is interesting, but rather technical, not very accessible to a generic audience and of interest for specific readers. Also the presented experiment is of interest and sound, but rather standard. For these reasons, I suggest publishing it in a more specific journal.

We were pleased to hear that the referee considers our present work interesting. Indeed, we believe that the central finding of our paper – namely, that there can be quantum coherence between two causal relations, which is qualitatively different from conventional superpositions of quantum states – is a fundamental new insight about the interplay of quantum theory and causality. This should be of particular interest to the emerging field of quantum causality, which bridges the areas of quantum gravity and foundations. However, considering the central role that causal explanations play in all disciplines of science, this result should also be of interest to a much broader readership, and as such would be well placed in Nature Communications.

The classification of different combinations of causal relations does indeed require a certain level of mathematical rigor, and in the supplementary material we aim to give a sufficiently thorough technical account that will satisfy an expert in the field. However, we believe that the presentation in the main paper is straightforward and makes the point of our work accessible to a non-expert audience. (Indeed, none of the other reviewers seemed to consider this a problem.) In order to further reduce the burden on the reader, we opted to present the development of the relevant theory together with our experimental findings, so that readers are not required to be familiar with or consult a second paper on the topic in order to understand our results.

The experimental component of our work is in fact only the second realization of causal tomography (a scheme, first proposed in 2015, which generalizes conventional quantum state and process tomography in order to characterize causal relations). We have made significant technical improvements to the linear optics implementation of the partial swap, which is a useful gate for a variety of applications (PRA 80 062306, PRL 88 097905), simplifying its construction and making it more robust to noise.

Reviewer #2:

The manuscript deals with various kinds of causal relations that interpolate between common cause and cause-effect. The authors distinguish classical and “quantum” causal relations, introducing notions of “quantum common cause” (two quantum systems in an entangled state) and “quantum-cause-effect relation” (one quantum system obtained from another through a coherent quantum transformation). Moreover, the authors provide criteria to distinguish different types of causal relationships and illustrate the distinction in a photonic experiment.

The general topic is interesting and the manuscript is nicely written. However, my opinion is that the degree of innovation is not sufficient for publication in Nature

Communications. Characterizing quantum states and quantum maps in relation to entanglement is a standard approach and a large part of the manuscript is just a translation/application of existing notions to the topic of causal relations.

While such translation can be interesting, the authors seem to have gone a bit too far when using new names like "causal maps" and "causal tomography" for already well established notions. The theory of witnesses is also well understood and its application to the causal scenario is relatively simple. The properties of the partial swap are well-known in quantum computing and the experimental implementation of this two-qubit gate has become standard by now.

Reviewer #2 points out that some of the notions we use in the present manuscript can be identified with concepts from previous work, for example that coherence in the common-cause and cause-effect paths correspond to the familiar categories of entangled bipartite states and entanglement-preserving quantum channels, respectively. However, we only use these established components as references in presenting our central innovation: the possibility of coherence in the way that common-cause and cause-effect relations are *combined*.

A combination of these two causal relations cannot be represented by either bipartite quantum states or quantum channels, requiring instead a new type of mathematical representation, termed causal map or quantum comb (PRA 80 022339). The task of characterizing these combined causal relations prompted the development of the causal tomography scheme, which was introduced in 2015 and combines aspects of state and process tomography, but with the ultimate goal of discerning causal relations.

Along similar lines, we agree that the use of entanglement as an indicator of coherence in a bipartite state is certainly conventional, and using entanglement of the bipartite Choi state of a channel to detect the fact that the channel preserves some coherence is also a well-established technique. The recognition that a bipartite state corresponds to a common-cause connection while a channel corresponds to a cause-effect connection is then a small step. However, the sort of channel to which we apply entanglement as an indicator of coherence is of an entirely novel sort. It is one where the input (denoted D in our article) is later in time than the output (denoted C in our article), but where one is post-selecting on a variable that is a common effect of D and some cause of C . Without the post-selection, this channel would generate no correlations between D and C , let alone any coherence. Furthermore, this sort of entanglement, which we termed the Quantum Berkson Effect, is leveraged as an indicator of coherence between two causal relations, which had not been considered before.

We agree with the referee that the properties of the partial swap are well established. However, the goal of our experiment was not to study these but rather to use the partial swap as a way of connecting two quantum variables via different causal structures. Concerning its experimental implementation, the working principle was based on the implementation by Cernoich *et al.* (Ref. [23]). However, due to the large number of measurements required to reconstruct the causal map for each class of interest, our experiment required a gate that was stable over multiple measurements. We significantly improved the stability and fidelity of the 2-qubit gate by replacing the balanced Mach-Zehnder interferometer in the original implementation with a folded Sagnac interferometer. This also removed the need for active phase stabilization. To our knowledge, this is the first optical implementation of the partial swap with these passive

stability properties, which may be useful in future experiments.

Another aspect that I find unsatisfactory is the motivation. The authors motivate the study of quantum causality in general, but only provide vague motivations for the specific work done in this manuscript. Quantum gravity, foundational puzzles, and quantum technologies are briefly mentioned, but without any direct connection with the results.

We have sought to clarify why the specific work in this article is likely to have relevance to the fields we noted. In particular, we have emphasized the pre-existing focus on superpositions of causal structures in the quantum gravity literature, to which our work is directly relevant, and the fact that our results should lead to a novel classification of types of non-Markovian dynamics in quantum theory. The motivation given in the introduction of the manuscript has been modified to read as follows:

“Considering the central importance of causal explanations in science, the fact that causal relations can exhibit a typically quantum behaviour is striking. Moreover, causal structure is a close proxy for the structure of space-time in general relativity, and it has been suggested that we will have to abandon the notion of definite causal structure and instead allow superpositions thereof in order to develop a theory of quantum gravity [8,9]. The present work provides a framework and tools for doing so. At the same time, our program may provide new resources for practical applications: quantum causal relations have been proposed as a resource for computational tasks [10], with striking applications in gate discrimination [11-13].”

Furthermore, because non-Markovianity in open-system dynamics can be understood as arising from the environment acting as a common cause of the system at different times, our recognition of the fact that there are many ways of combining common-cause and cause-effect relations is likely to translate into tools for analyzing non-Markovianity. We have thus added a discussion on the topic of non-Markovianity in the concluding remarks.

Finally, two points in the manuscript are relatively weak. First, basing the definition of quantum causal relations on entanglement is one possibility, but not the only choice. "Entanglement" is just a specific instance of "quantumness" or "coherence", although the authors use these terms interchangeably.

Regarding our choice of the indicator of 'quantumness', we would first like to stress that we only established a *sufficient* condition – we do not claim to introduce a criterion that is both necessary and sufficient. We have attempted to make this distinction clear in the paper. Furthermore, while we agree that there are a number of possible indicators that one could use in this context, we felt that entanglement was preferable to the alternatives, for instance discord or Bell inequality violations, on the following grounds. Firstly, a good notion of quantumness is one that underlies resources, and the importance of entanglement as a resource (for instance for information processing and cryptography) is well established. Bell inequality violations also meet this requirement, but turn out to be too restrictive for the present application, since we can show that our scenario does not violate a Bell inequality. Entanglement is therefore an appropriate indicator for the non-classical effects realized in our experiment.

Second, the authors define the "quantum Berkson effect" as the conditional generation of entangled states through post selection. But Berkson's paradox is not only the observation that post selection can induce correlations between two uncorrelated variables. Taking this straightforward observation and replacing classical with quantum correlations (however defined) is fine, but it is not enough to make an interesting quantum version of Berkson's paradox.

We don't know what the referee has in mind when they claim that Berkson's paradox is "not only" postselection-induced correlations between uncorrelated variables. This appears to be the standard definition of the paradox. For instance, Judea Pearl, in his influential textbook on causality, defines the phenomenon as follows: "(Observations) on a common consequence of two independent causes tend to render those causes dependent, because information about one of the causes tends to make the other more or less likely, given that the consequence has occurred. This pattern is known as selection bias or Berkson's paradox (...)"

The referee questions whether our quantum version of Berkson's paradox is sufficiently "interesting". However, whatever intrinsic interest it might have is secondary, because it was introduced in our article merely for the purpose of illustrating the intuition behind our definition of a coherent mixture of causal relations. Berkson's paradox can be used in classical causal inference to distinguish between physical and probabilistic mixtures of cause-effect and common-cause relations, and therefore it is natural to define a *coherent* physical mixture as one wherein the induced Berkson correlations are quantum. In other words, our objective was not to simply define a quantum version of Berkson's paradox, but to use this phenomenon to define and witness a coherent mixture of causal relations. Therefore, we don't see how this is a weakness of the paper.

Reviewer #3:

The paper investigates the possible causal relations that can exist between two time-ordered quantum systems. The authors distinguish 4 classes of causal relations, whether the systems are linked by classical or quantum pathways, and by probabilistic or "physical" mixtures of common-cause and cause-effect relations. Inside the class 'PHYSQ' (physical mixture with quantum pathways), the additional subclass 'COH' is identified, where the systems exhibit a quantum version of the Berkson paradox. The authors construct witnesses to test which class the causal relations between a pair of systems belongs to. An experiment is carried out, where variations of the same setup allow the authors to demonstrate all different classes of quantum causal relations they identified.

The study of quantum causal relations has gained a lot of interest lately in the field of quantum foundations. The in-depth analysis of the various causal relations presented in the paper is thus particularly timely and interesting. The paper contains a wealth of information, is clearly written and well-organised. The data analysis in the experiment is well-described and appropriate.

There is just one aspect in the derivation of a witness for "physical mixtures" which did

not convince me, as I explain below. Before I can recommend the paper for publication, I would need the authors to address that point.

I also make below a few remarks and suggestions that the authors may want to consider to further improve the clarity of the paper, and list a few typos in some equations of the Supplementary Information that I spotted.

**** Witness of physical mixture ****

After Eq.(S65) of the Supplementary Information, and in the following derivation, it seems to be implicitly assumed that $\text{tr}_B \rho_{CB}$ from the first term of (S65) (corresponding to the common-cause case) and ρ_C from the second term (corresponding to the cause-effect case) are the same. I do not see why this should necessarily be the case: couldn't we consider a probabilistic mixture of common-cause and cause-effect relations, for which the restrictions to the system C do not coincide?

Consider for instance the following 2 (classical) causal maps:

$$P_{ce}(cbd) = P_{ce}(c) P_{ce}(b|d) \quad u(d) = \frac{1}{2} \delta_{\{c,+1\}} \delta_{\{b,d\}}$$

$$P_{cc}(cbd) = P_{cc}(c) P_{cc}(b|c) \quad u(d) = \frac{1}{2} \delta_{\{c,-1\}} P_{cc}(b)$$

Clearly, P_{ce} and P_{cc} are of the cause-effect and common-cause forms, respectively.

*Define now the probabilistic mixture $P = 1/2 * P_{ce} + 1/2 * P_{cc}$. Calculating the witness C_{CD} of Eq.(S77), I find $C_{CD} = 1/2$ (the same value as theoretically predicted for the experimental implementation of PHYSC and COH), in contradiction with the claim that $C_{CD} = 0$ for all probabilistic mixtures of cause-effect and common-cause pathways.*

It thus seems to me that the witness proposed by the authors to identify physical mixtures of cause-effect and common-cause pathways is not valid in general, without the additional unjustified (?) assumption that " ρ_C " is the same for both pathways. Another witness would need to be constructed and measured to make the experiment conclusive in this regard.

(Another argument can indeed be given to show that the observed correlations in the PHYSC and PHYSQ experiment are physical mixtures.)

Or did I misunderstand something?

In the first section of the supplementary material, we noted explicitly that "in the case of a probabilistic mixture of common-cause and cause-effect relations between A and B, the variable that controls which term in the mixture is realized must act only on B." We go on to explain that the reason for this restriction is that were it not made, then the control variable itself would act as a common cause of A and B. This is a problem because a probabilistic mixture of purely cause-effect maps would then not be purely cause-effect. The restriction implies that the marginal on C cannot depend on the control variable and consequently that this marginal must be the same in every term of a probabilistic mixture. As such, the example the referee has provided is not a valid probabilistic mixture of cause-effect and common-cause relations according to our definition, but rather a physical mixture, and consequently it is appropriate that our witness of physical mixture has a nonzero value for this example.

Looking over what we wrote, however, we noted that this subtlety of the definition of a probabilistic mixture was not made explicit in the discussion provided in the main text. Indeed, the definition provided in the main text was deficient insofar as it failed to mention the restriction on ρ_C . We have consequently revised the discussion in the main text to read as follows:

“A causal map is said to be a *probabilistic mixture* of cause-effect and common-cause relations if there is a hidden classical control variable, J , which influences only B such that for every value of J , either B depends only on D in the causal map or B depends only on its common cause with C . We show in the Supplementary Information that every such causal map can be expressed as having just one term of each type, $\mathcal{E}_{CB|D} = w \mathcal{E}_{B|D} \otimes \rho_C + (1-w) \rho_{CB} \otimes \text{Tr}_D$, where $0 \leq w \leq 1$ and where $\text{Tr}_B \rho_{CB} = \rho_C$. The fact that the marginal on C is the same in both terms follows from demanding that the control variable J does not influence C . This demand is justified as follows. A probabilistic mixture of causal maps that are all purely cause-effect should also be purely cause-effect. But if J could influence C in addition to B , then it would itself constitute a common cause of A and B , and the requirement would be violated.”

Furthermore, we have rewritten the first section of the supplementary material to clarify this issue. It now addresses the quantum case rather than the classical case, provides a simplified derivation of the fact that the causal map reduces to just two terms, emphasizes that the marginal on C is the same in both terms and discusses why the proposed definition, wherein the control variable only has an influence on B , is the only sensible definition of such a probabilistic mixture. In particular, we note that allowing the control variable to influence C and B would have the consequence that a probabilistic mixture of purely cause-effect maps would not itself be purely cause-effect, and that a probabilistic mixture of causally-disconnected maps would not itself be causally disconnected.

**** Other remarks / questions ****

Main text:

- On the choice of terminology: to me, what the authors call a “coherent mixture” would be a quantum superposition; to refer to what they call a “physical mixture” of cause-effect and direct cause, I would have talked about a “combination” of cause-effect and direct cause.

I was wondering if there was a specific reason, why the authors have chosen the same terminology of “mixture” for all situations? Some comment may be useful.

(It seems to me that “superposition” is more common in the literature, while it’s not used at all in the paper. Is “physical mixture” a standard terminology, or one that is introduced by the authors?)

We first address the question of why we used “physical mixture” rather than “combination”. The reason is simply that probabilistic mixtures are not the only sort of mixtures. Indeed, they are the exception. A drink that is perhaps purely orange juice and perhaps purely vodka, with equal probability, is a probabilistic mixture of orange juice and vodka. A drink that is half orange juice and half vodka is a physical mixture. In our article, it is important that the entities which

we are mixing -- cause-effect and common-cause relations -- are not mutually exclusive possibilities. One need not have just one or just the other, but rather, like a mixed drink, both may be present. We felt that it was important to emphasize this fact and our terminology helps to do that. We are introducing this terminology in our article, as is clear from the fact that we say “*We refer to this* as a physical (as opposed to probabilistic) mixture of the two mechanisms.”

Regarding “coherent mixture” versus “superposition”, the notion of superposition has a more limited scope of applicability than that of coherent mixture: only if the entities one is combining are represented by *pure states* can one speak of superpositions. For instance, given a distinguished basis of Hilbert space, the only states that can properly be said to be in a superposition of basis states are pure states. If the density operator merely contains off-diagonal elements relative to this basis, it is typically said to exhibit coherence relative to it. The notion of coherence is the more general concept and it is therefore the one we that we opted to use in our article.

- p.3, l.51, “realizing this possibility requires exotic new physics”: note that the “quantum switch” has been realised in a rather standard photonic circuit.

The experiment of Procopio et al. has not in fact realized a superposition of causal orders. We have added a section to the supplementary material to explain why this is the case and to justify the claim that doing so is likely to require exotic physics.

- p.9, l.122, why is it required that τ_{CD}^b is entangled for all b ? Why would it not be enough that there just exists one b , for which it is entangled? Is there a counter example? Some comment here would be useful. Same questions for τ_{CB}^d and τ_{BD}^c on p.10.

This is why we describe our condition as merely sufficient. One *might* want to only require quantumness for the correlations induced by post-selecting on one value rather than requiring this for the correlations under every post-selected value. Anyone who endorses the weaker requirement as sufficient must endorse the stronger requirement as well. Our choice is therefore most able to satisfy a sceptic.

- p.9, l.131-132: the result is proven in the Supplementary Material for the classical case; some argument (or at least some quick comment) should be given to justify that it also applies to the quantum case.

As noted in the response to the referee’s question about the witness of physical mixture, we have rewritten this section of the Supplementary Material to deal explicitly with the quantum case.

- p.10, l.137, why is it a sufficient condition, and not a necessary one? Again, some clarification would be useful.

In this article, our notion of quantumness for causal pathways and the Berkson effect is derived from a notion of quantumness for induced correlations associated with these. We opted to apply a condition on correlations that all would agree is sufficient for judging them to be quantum, namely, entanglement. While some might claim certain types of unentangled correlations to be quantum as well, and hence to argue that our condition is not necessary, we

avoid such a debate by only claiming sufficiency of the condition.

- The definition of the class COH requires that (i) the causal map is quantum in both pathways and (ii) it exhibits a Quantum Berkson Effect (QBE). This suggests that one may have a QBE outside of the class PHYSQ? Where else would that be possible, and what would it look like?

It is indeed possible to construct an example that exhibits the QBE yet is classical in both pathways, so that it falls into the class PhysC. To wit, consider the partial swap gate, which serves as an example of Coh in the article, followed by complete dephasing in the σ_z eigenbasis on B. The dephasing does not interfere with the QBE, since observing the QBE requires one to post-select on σ_z eigenstates on B anyway. However, complete dephasing on B does entail that the causal map is classical in both the common-cause and cause-effect pathways, since for all measurements on C, the induced states on BD are separable, and for all preparations on D, the induced states on CB are separable.

- In the classification of the different classes of mixtures: I guess we could have situations where only one pathway (cause-effect or common cause) is quantum, the other one being classical? That would be situations that fit in none of the various classes. Would the authors like to comment on that?

The referee is correct in pointing out that one can have quantumness in one pathway but not another. It was merely for the sake of brevity that we focused on the cases that are either fully classical or 'fully' quantum. We now point out in the caption of figure 3 that “We leave aside cases wherein only one pathway is quantum.”

- Figure 4: I find it rather confusing that the red lines connect C and D. I would suggest to draw them as in Fig. 2.

We have changed the figure as suggested.

-In the caption, 3rd line, by “equal mixture” I guess one means “equal probabilistic mixture”? 4th line, one may clarify in which basis the “complete dephasing operations” are performed (or say that different bases are considered).

We have modified the caption of figure Fig. 4 to read “equal probabilistic mixture”.

- p.19, l.240: it may be misunderstood that only preparations and measurements in the $\{H,V\}$ basis are required to evaluate the negativities. One may clarify here that a full topography is actually needed (as described in the Suppl. Info.).

We have modified the passage to read “The indicators \mathcal{N}^c_{BD} , \mathcal{N}^d_{CB} and \mathcal{N}^b_{CD} are evaluated using tomographic reconstructions of the operators τ^c_{BD} , τ^d_{CB} and τ^b_{CD} , respectively, under preparations (on $\$D\$$) and measurements (on $\$C\$$ and $\$B\$$) of $\{|\ket{H}\rangle, |\ket{V}\rangle\}$.”

Further clarification is given just one short paragraph above, where we point out that “We reconstruct the operators τ^c_{BD} , τ^d_{CB} and τ^b_{CD} using subsets of tomographic data (for example, using only runs that found $\$B\$$ in the state \ket{H} to reconstruct τ^H_{CD} .” Between these two statements, we hope to have made our

method clear.

Supplementary Information:

- p.1, is E really needed to mediate the influence of λ ? Couldn't one simplify the analysis and let λ directly influence B?

We have rewritten the first section in the supplementary information and no longer use the intermediary variable E as suggested.

- In "Suppose that the conditional $P(B|DEJ)$ reduces to $P(B|DJ)$ for all values of J", I guess that should also be for all values of D?

We have made the suggested simplification and clarification in the process of rewriting the first section of the supplementary information.

- Just before Eq. (S8), it may be useful to recall that ρ_{CE} will always be taken to be $|\Phi^+\rangle$.

We have added the clarification "(...) ρ_{CE} ", which will be taken to be the maximally entangled state $|\Phi^+\rangle$."

- In the paragraph that follows Eq. (S17), it may be useful to refer to Eq. (S12) to clearly see which "two-qubit identity operator" and "swap operator" one is referring to.

We have added a reference pointing to the equation suggested by the referee.

- In Eq.(S32), $|c\rangle\langle c|$ is not defined. It may be useful also to say how it is linked to $|e\rangle\langle e|$, and clarify how $P(cb|d)$ is obtained from $P(bf|de)$.

We have added the following clarification: "The dephasing on E also effectively reduces C to a classical binary variable, since C is only related to other variables in the problem via E. We denote this variable by c and the corresponding preferred basis (which generally depends on the initial joint state CE) by $|\ket{c}\rangle$."

- It seems that Eq. (S33) assumes that $\{|d\rangle\}$ is the basis of the Choi-isomorphism? But this is not the case in the analysis that follows, where the Y basis will be used (cf e.g. Eq. (S53)). This should be clarified.

We thank the referee for raising this question, which turned out to be somewhat subtle and to have implications for the apparent sign inconsistencies in other parts of the manuscript. If the Choi isomorphism was defined with respect to the basis $\{|d\rangle\}$, then Eq.(S33), as given in the original manuscript, would be compatible with Eq.(S32). However, we take $\{|d\rangle\}$ to be the basis with respect to which D is dephased, whereas the basis of the Choi isomorphism is fixed to be $\{H,V\}$ throughout our work. It follows that we must modify Eq.(S33), writing the Choi state in terms of a new basis $|\ket{\tilde{d}}\rangle$, which "is related to $\{|d\rangle\}$ by complex conjugation in the basis that defines the Choi isomorphism". This change also has implications for Eq.(S53), where it helped us resolve the sign issues that the referee had pointed out.

- It may be useful also to clarify that Eq.(S34) is obtained for $\rho_d = |d\rangle\langle d|$ and $|\Phi_{C^c} = |c\rangle\langle c|$.

We have added clarification before Eq.(S34).

- Sometimes the indices c, b, d are taken to be $0, 1$; sometimes they are taken to be ± 1 . It may be useful to clarify this, and when the authors switch from one to the other.

We thank the referee for pointing out this inconsistency. We believe we have now eliminated all usage of the values $\{0, 1\}$. In order to further reduce the risk of confusion, we have eliminated the symbol \oplus , which conventionally denotes the XOR operation between two binary variables with values $\{0, 1\}$, and now take instead the product of the variables in question, which is a more natural representation of the XOR for values $\{+1, -1\}$.

- Top of p.6, 2nd line, I understand that the result is proven only if c and d have uniform prior distributions?

We have clarified this point on p.6 and modified the general proof in the last section accordingly: it now proceeds for the most part without any assumptions on the priors and only adds the assumption of uniform priors when we want to explicitly evaluate the upper bound on the mutual information.

*- Bottom left of p.6, in "However, the causal map cannot be realized by a probabilistic mixture of purely common-cause and purely cause-effect relations, since the last term sets $b = c \oplus d$ ".
Is that argument really enough? Couldn't one imagine a more complicated probabilistic mixture of common-cause and cause-effect relations, where the two terms combine in such a way that the relation $b = c \oplus d$ appears in the combination, while it would be neither in one term, nor in the other? I feel that some stronger argument should be given here, to explain why this cannot be the case.*

We have modified the argument to read as follows:

"However, the causal map cannot be realized by a probabilistic mixture of purely common-cause and purely cause-effect relations, since the witness of physical mixture is $\epsilon/4$, i.e. non-zero for all valid values of ϵ . (The fact that the last term, $b = -cd$, has b depending simultaneously on c and d is also suggestive of a physical mixture, but not conclusive.)"

Indeed, one could construct the following (somewhat contrived) counter-example: an equal probabilistic mixture of one term that sets $b = cd$ and another which sets $b = -cd$, generates a total probability distribution wherein all variables are uncorrelated.

- Bottom left of p.7, it may be useful to say that the map $\{cal E\}_{BF|DE}$ is modified from the original (S45), not from (S48).

We have added this clarification.

***** A few typos in the Supplementary Information *****

- Eqs. (S15-S21), there are some $1/\sqrt{2}$ and $1/2$ factors missing (where exactly they are missing depends on how the authors want to define A_k and B_k).

- Eq. (S25), I believe there is a d_B factor missing (as in (S7)). It then sounds a bit weird to

read, just before, that “[τ_{CB^d}] does not require renormalization to ensure unit trace”.

- There is a 1/2 factor missing in Eq. (S29) (which should be consistent with (S22)).

- I believe there is a 1/2 factor missing in Eq. (S50).

We have addressed all of these mistakes as suggested and thank the referee for their thoroughness in pointing them out.

After Eq. (S25), we have changed the text to read 'the renormalization factor to ensure unit trace is always $\$d_D\$$ '.

- Eq. (S53), I found the same 4 terms, but with all $\pm y$ states flipped (so that $b \oplus c \oplus d = 0$, rather than 1).

We agree and thank the referee for pointing this out. We have revised Eq. S53 as well as the associated classical probability distributions.

Am I correct also to think that $P(cb|d)$ in that equation is not the same as in the definition of the maps (e.g. from Fig.4), but differs by a flip of d (due to its measurement in the Y basis, and the partial transpose being applied somewhere...)?

We have changed the example circuits for PhysC for the XNOR to match what was implemented in the lab.

- Eq. (S75), there should be a partial transpose on system D . (And putting it last may look more natural?)

We have added the partial transpose and changed the order, since we agree that this is more natural.

Reviewer #4:

The authors theoretically describe and experimentally demonstrate various classes of causal structures that can occur in a two-qubit quantum system. This is an interesting and timely result in the novel field of quantum causality, which has led to important recent developments in quantum foundations.

Although in parts quite dense, the manuscript is well written and sufficiently introduces the studied concepts and the experimental design. The Supplementary Information is very comprehensive on both the experimental and theoretical aspects of the work and complements the paper. However, I feel that the excellent technical presentation comes somewhat at the cost of the discussion of the physical implications and related work. I have made a few specific suggestions below.

Overall the quality of the manuscript is very high and upon extension of the discussion, I can recommend publication in Nature Communications.

1) In the second paragraph on page 3 the authors mention that the present work is based on Ref.[3,13]. Of these Ref.[3], which shares many of the present authors, uses a similar experimental setup to study cause-effect and common-cause structures in

quantum mechanics. I would recommend to expand on the relation between the current work and Ref.[3].

We have expanded on the relation between this work and Ref [3] by adding:

“This latter circuit realizes the same causal map implemented in Ref. [3], which focused on the task of resolving probabilistic mixtures of cause-effect and common-cause relations. However, the experimental setup of Ref. [3] could not realize physical mixtures, which are required to address the broader question, investigated in the present work, of how these two extremes may be combined in general. “

2) The authors show in the supplementary information that the example from the class PHYSC can be realized by implementing dephasing in different bases. What is the physical intuition behind the choice/requirement for different bases?

We have added the following clarification to the supplementary information: “(...) by applying complete dephasing in the eigenbases of σ_x on EE , σ_y on ED and σ_z on EB . This choice of bases ensures that the witness of physical mixture, which is evaluated using only measurements of these particular observables, remains unchanged by the dephasing. It therefore ensures that we continue to realize a physical mixture while eliminating the coherence in the cause-effect and common-cause paths.”

3) It would be interesting to discuss the power of the introduced witness of physical mixture. The authors show that it is zero for all probabilistic mixtures, but it is not discussed what fraction of physical mixtures it can detect.

The referee raises an interesting question. While one can find certain families of causal maps for which our witness does reveal the presence of physical mixture, one can construct other examples of physical mixtures for which this is not the case. It is unknown whether, for every causal map that realizes a physical mixture, there exists some witness that reveals this fact. In order to address the question raised by the referee, one would first have to characterize the set of all physical mixtures of common-cause and cause-effect relations, which may be a non-trivial problem in itself. Furthermore, we note that the witness of physical mixture which we propose is mathematically rather different from conventional witnesses of entanglement (for instance, it is not linear in the causal map), so that existing results and techniques for determining the power of entanglement witnesses will generally not carry over. In light of these considerations, we feel that the question raised by the referee would exceed the scope of our present work, but would rather make an interesting starting point for future work.

4) There seems to be a typo in the caption of Fig.6, where left-circular is denoted $|R\rangle$ and right-circular $|L\rangle$, respectively.

We have corrected the mistake and thank the referee for pointing it out.

5) The discussion section focuses mainly on the case of multiple mechanisms, but is missing a discussion of the physical implications of coherent mixtures of causal mechanisms. Specifically, it would be interesting to discuss what the discovery of such situations contributes to our understanding of quantum causality or any potential applications as in the case of a superposition of causal orders.

The referee raises an important point concerning the discussion. We have reformulated the discussion to highlight the contributions we feel this work brings to the field of quantum causality and in addition, we have added a paragraph which describes the implications of quantum-coherent causal structures to the topic of non-Markovianity.

Reviewer #2 (Remarks to the Author):

The revised manuscript has substantially improved, addressing most of the comments in my previous report. My overall assessment of the revision is largely positive, both on the content of the work and on its suitability for publication in Nature Communications.

In relation to the authors' response, I would like to follow up on a few points, including some where I hold that improvements are still needed.

1) "We only use these established components as references in presenting our central innovation: the possibility of coherence in the way that common-cause and cause-effect relations are combined".

I see the point and agree.

2) "The task of characterizing these combined causal relations prompted the development of the causal tomography scheme, which was introduced in 2015 and combines aspects of state and process tomography, but with the ultimate goal of discerning causal relations."

I agree that the expression "causal tomography" was introduced in the 2015 work referenced in the manuscript. My question, however, is whether this notion is conceptually different from the notion of tomography of multitime processes, discussed in some of the earlier papers referenced by the authors.

The two notions seem to coincide, but perhaps there are some nuances I am missing. Either ways, this point should be clarified.

3) "We have sought to clarify why the specific work in this article is likely to have relevance to the fields we noted. In particular, we have emphasized the pre-existing focus on superpositions of causal structures in the quantum gravity literature, to which our work is directly relevant, and the fact that our results should lead to a novel classification of types of non-Markovian dynamics in quantum theory."

I appreciate the improvement in the presentation on both fronts. I think that the positioning of this work with respect to the previous literature is more balanced now. I still find a bit curious that, when the notion of causal map is first introduced, the sole reference is the authors' 2015 paper.

4) "Regarding our choice of the indicator of 'quantumness', we would first like to stress that we only established a sufficient condition – we do not claim to introduce a criterion that is both necessary and sufficient."

This clarification is an important improvement, indeed.

On a minor point, I would suggest to similarly clarify the use of the term "coherence preserving". The maps considered in the paper are actually "entanglement preserving"—a proper subset of the set of coherence preserving maps.

5) "We don't know what the referee has in mind when they claim that Berkson's paradox is "not only" postselection-induced correlations between uncorrelated variables. This appears to be the standard definition of the paradox."

I wouldn't say that it is the standard definition, but rather the standard generalization of the paradox. The generalization, per se, has no paradoxical feature. For example, knowing that the sum of two dices is 11 generates a correlation, whereby the dices can only have values (5,6) and (6,5), even if they were rolled independently. Postselection induces correlations, but not always with counterintuitive features.

The abstract advertises a quantum version of Berkson *paradox*, which however is not present in the paper. The body of the paper uses the wording "quantum Berkson effect", which I find more accurate. My recommendation is to change the abstract, replacing "quantum version of Berkson's paradox" with "quantum version of Berkson's effect, whereby two independent causes can become correlated upon observation of their effect."

Reviewer #3 (Remarks to the Author):

The authors have satisfyingly addressed all the points I raised in my previous report.

Even if certain aspects of the work may seem to be simply rewordings of standard notions in quantum information (as other reviewers have pointed out), I still believe that the clarification of concepts presented in the paper make it quite interesting and significant for the study of quantum causal relations. I am happy to now recommend it for publication in Nature Communications.

Reviewer #4 (Remarks to the Author):

I believe that all reviewer comments, including my own, have been sufficiently addressed and appropriate changes have been made to the manuscript. The authors make a good case for the relevance of their work and extended the motivation and discussion sections, which improves the appeal to the broad readership of Nature Communications.

The manuscript remains technical and dense in parts (e.g. p8-p10), however, it is well written and should be understandable to a general physics audience. The work remains timely in light of the current developments in quantum foundations and the study of causality, and I can recommend the manuscript for publication in Nature Communications.

Reviewer #2:

The revised manuscript has substantially improved, addressing most of the comments in my previous report. My overall assessment of the revision is largely positive, both on the content of the work and on its suitability for publication in Nature Communications.

In relation to the authors' response, I would like to follow up on a few points, including some where I hold that improvements are still needed.

1) *"We only use these established components as references in presenting our central innovation: the possibility of coherence in the way that common-cause and cause-effect relations are combined".*

I see the point and agree.

2) *"The task of characterizing these combined causal relations prompted the development of the causal tomography scheme, which was introduced in 2015 and combines aspects of state and process tomography, but with the ultimate goal of discerning causal relations."*

I agree that the expression "causal tomography" was introduced in the 2015 work referenced in the manuscript. My question, however, is whether this notion is conceptually different from the notion of tomography of multitime processes, discussed in some of the earlier papers referenced by the authors.

The two notions seem to coincide, but perhaps there are some nuances I am missing. Either ways, this point should be clarified.

We acknowledge that the notion of tomography from Silva *et al.* (PRA 89 012121) or tomography of quantum combs (PRA 80 022339, arXiv:1601.04864) would be adequate for our purposes. For this reason, we have replaced "causal tomography" when it is introduced with "tomography" and have added the additional references. We thank the Referee for pointing this out.

The section now reads,

"We characterize the causal maps realized in the experiment using tomography^{9,12,16,24}"

3) *"We have sought to clarify why the specific work in this article is likely to have relevance to the fields we noted. In particular, we have emphasized the pre-existing focus on superpositions of causal structures in the quantum gravity literature, to which our work is directly relevant, and the fact that our results should lead to a novel classification of types of non-Markovian dynamics in quantum theory."*

I appreciate the improvement in the presentation on both fronts. I think that the positioning of this work with respect to the previous literature is more balanced now. I still find a bit curious that, when the notion of causal map is first introduced, the sole reference is the authors' 2015 paper.

We agree that the previous version of the manuscript was not clear about the fact that many other formalisms in the literature would be adequate to describing our experiment. We previously provided a comprehensive list of citations to such formalisms in the introduction. We have modified the section where we introduce the causal map to point out that one can infer how our circuit ought to be represented as a CP map from *any* of these references,

“Mathematically, the circuit’s functionality is represented by a trace-preserving, completely positive map from states on D to states on the composite CB [...] as can be inferred from Refs. 9,12–14,16 , and which we term a causal map.”

We have also modified the introduction:

“quantum-coherent mixtures can only be represented using recent extensions of the formalism¹⁰⁻¹⁶, in particular Refs. 9,17 .”

4) “Regarding our choice of the indicator of 'quantumness', we would first like to stress that we only established a sufficient condition – we do not claim to introduce a criterion that is both necessary and sufficient.”

*This clarification is an important improvement, indeed.
On a minor point, I would suggest to similarly clarify the use of the term “coherence preserving”. The maps considered in the paper are actually “entanglement preserving”—a proper subset of the set of coherence preserving maps.*

We have clarified this point in the paper by changing instances of “coherence-preserving” with the term “not entanglement-breaking”. We thank the referee for this suggestion.

5) “We don't know what the referee has in mind when they claim that Berkson's paradox is "not only" postselection-induced correlations between uncorrelated variables. This appears to be the standard definition of the paradox.”

I wouldn't say that it is the standard definition, but rather the standard generalization of the paradox. The generalization, per se, has no paradoxical feature. For example, knowing that the sum of two dices is 11 generates a correlation, whereby the dices can only have values (5,6) and (6,5), even if they were rolled independently. Postselection induces correlations, but not always with counterintuitive features.

*The abstract advertises a quantum version of Berkson *paradox*, which however is not present in the paper. The body of the paper uses the wording “quantum Berkson effect”, which I find more accurate. My recommendation is to change the abstract, replacing “quantum version of Berkson’s paradox” with “quantum version of Berkson’s effect, whereby two independent causes can become correlated upon observation of their effect.”*

We have changed instances of “Berkson’s paradox” to “Berkson’s effect”. In addition, the description of Berkson’s effect in the abstract now reads,

“[...] a quantum version of Berkson’s effect, whereby two independent causes can become correlated upon observation of their common effect”.

We thank the Reviewer for their comments and suggestions during the review process

Reviewer #3:

The authors have satisfyingly addressed all the points I raised in my previous report.

Even if certain aspects of the work may seem to be simply rewordings of standard notions in quantum information (as other reviewers have pointed out), I still believe that the clarification of concepts presented in the paper make it quite interesting and significant for the study of quantum causal relations. I am happy to now recommend it for publication in Nature Communications.

We thank the Reviewer for their suggestions and comments during the review process.

Reviewer #4:

I believe that all reviewer comments, including my own, have been sufficiently addressed and appropriate changes have been made to the manuscript. The authors make a good case for the relevance of their work and extended the motivation and discussion sections, which improves the appeal to the broad readership of Nature Communications. The manuscript remains technical and dense in parts (e.g. p8-p10), however, it is well written and should be understandable to a general physics audience. The work remains timely in light of the current developments in quantum foundations and the study of causality, and I can recommend the manuscript for publication in Nature Communications.

We thank the Reviewer for their suggestions and comments during the review process.